# Tcf1$^+$ cells are required to maintain the inflationary T cell pool upon MCMV infection

Suzanne P. M. Welten [1], Alexander Yermanos[1,2,5], Nicolas S. Baumann[1,5], Franziska Wagen[1], Nathalie Oetiker[1], Ioana Sandu [1], Alessandro Pedrioli[1], Jennifer D. Oduro[3], Sai T. Reddy[2], Luka Cicin-Sain [3], Werner Held [4] & Annette Oxenius [1]✉

Cytomegalovirus-based vaccine vectors offer interesting opportunities for T cell-based vaccination purposes as CMV infection induces large numbers of functional effector-like cells that accumulate in peripheral tissues, a process termed memory inflation. Maintenance of high numbers of peripheral CD8 T cells requires continuous replenishment of the inflationary T cell pool. Here, we show that the inflationary T cell population contains a small subset of cells expressing the transcription factor Tcf1. These Tcf1$^+$ cells resemble central memory T cells and are proliferation competent. Upon sensing viral reactivation events, Tcf1$^+$ cells feed into the pool of peripheral Tcf1$^-$ cells and depletion of Tcf1$^+$ cells hampers memory inflation. TCR repertoires of Tcf1$^+$ and Tcf1$^-$ populations largely overlap, with the Tcf1$^+$ population showing higher clonal diversity. These data show that Tcf1$^+$ cells are necessary for sustaining the inflationary T cell response, and upholding this subset is likely critical for the success of CMV-based vaccination approaches.

[1] Institute of Microbiology, ETH Zürich, Vladimir-Prelog-Weg 4, 8093 Zürich, Switzerland. [2] Department of Biosystems and Engineering, ETH Zürich, Mattenstrasse 26, 4058 Basel, Switzerland. [3] Department of Vaccinology and Applied Microbiology, Helmholtz Centre for Infection Research, Hannover-Braunschweig Site, 38124 Braunschweig, Germany. [4] Department of Oncology, University of Lausanne, 1066 Epalinges, Switzerland. [5] These authors contributed equally: Alexander Yermanos, Nicolas S. Baumann. ✉email: aoxenius@micro.biol.ethz.ch

C ytomegalovirus (CMV) is a β-herpesvirus that is universally present in the world's population, though, this infection is largely asymptomatic in healthy individuals. Upon human CMV (HCMV) and murine CMV (MCMV) infection an atypical T cell response is initiated, characterized by an accumulation of functional effector-like virus-specific CD8 T cells in blood and peripheral tissues, a process termed "memory inflation"[1–4]. Although there are CMV-specific T cells that follow the classical pattern of expansion, contraction and formation of long-term central memory pools ($T_{CM}$: CD62L+/CD127+/KLRG1−/CD27+), during viral latency the immune response is dominated by inflationary T cell responses that are restricted to a few epitopes[5]. These cells have an activated effector memory/effector-like phenotype ($T_{EM}$: CD62L−/CD127−/KLRG1+/CD27low), suggestive of repetitive antigen encounter. One major factor driving T cell inflation is the recurrent presentation of viral antigen due to the ability of the virus to establish long-term latency and to sporadically reactivate. While active viral replication is generally not measurable during viral latency, viral peptides that are processed by the constitutive proteasome[6,7] are likely present and are presented by latently infected non-hematopoietic cells[8,9].

Due to the large numbers of effector-like T cells in peripheral tissues, the use of CMV-based vectors has gained interest for vaccination purposes. In animal models, high numbers of functional CD8 T cells against foreign introduced antigens are induced with these vectors and these cells provide protection against heterologous viral and tumour challenges expressing these antigens[10–16]. Although one has to be careful with using live viral vectors for vaccination purposes, attenuated CMV strains that are spread-deficient also induce memory inflation, providing a safer alternative[17]. Furthermore, some replication incompetent adenoviral vectors elicit inflationary T cell responses in mouse models that show similarities with MCMV-induced inflationary T cells[18,19]. As the success of CMV-based vaccines is based on the induction of large populations of effector-like CD8 T cells in peripheral tissues[2], and the protective capacity of these vectors is directly correlated to the size of the peripheral T cell pool[10,11,20], it is important to delineate the factors that maintain this population at high numbers.

Although the half-life of inflationary T cells in mice is estimated to be around 6–8 weeks in circulation and 10–12 weeks in the periphery[21,22], the peripheral pool of inflationary T cell reaches high numbers and eventually stabilizes. This implies that there is continuous replenishment of the peripheral effector cells[23]. A small subset of the inflationary MCMV-specific T cell population, enriched in the lymph nodes (LNs), has a $T_{CM}$ phenotype[8], judged by high expression of the LN homing receptor CD62L and enhanced proliferation capacity[24]. Adoptive transfer of MCMV-specific $T_{EM}$ or $T_{CM}$ cells into hosts latently infected with MCMV revealed that specifically the $T_{CM}$ cells responded to viral reactivation events by proliferation[8]. Expression of the transcription factor T cell factor 1 (Tcf1, encoded by Tcf7), is required for the formation of $T_{CM}$ cells[25,26]. Recently, an important role for Tcf1 was found in chronic lymphocytic choriomeningitis virus (LCMV) infection that induces T cell exhaustion, which is a state of T cell dysfunction characterized by a strong impairment of cytokine production and expression of co-inhibitory receptors. Tcf1 defines a T cell subset that has stem-cell features e.g. Tcf1+ cells maintain proliferative potential, have self-renew capacity and are able to produce more terminally differentiated T cells[27,28]. Although MCMV-specific CD8 T cells are not exhausted, differentiated effector-like T cells are continuously produced throughout infection. Here we determined whether upon MCMV infection Tcf1+ cells are critical for the maintenance of the inflationary T cell pool by fuelling the population of peripheral effector-like T cells.

We found that a small subset of inflationary T cells, enriched in LNs, expresses Tcf1. Specifically these Tcf1+ cells proliferate and give rise to Tcf1− effector-like cells required to maintain the peripheral inflationary T cell population.

## Results

**A subset of inflationary CD8 T cells expresses Tcf1.** We infected Tcf7GFP reporter mice expressing GFP (green fluorescent protein) under the control of the Tcf7 locus[27] with MCMV-Δm157 (referred to as MCMV) to determine Tcf1 expression kinetics in MCMV-specific CD8 T cells. M38-specific CD8 T cells followed the inflationary pattern, indicated by an accumulation in the blood, whereas the non-inflationary M45-specific CD8 T cells contracted after the acute phase of infection (Fig. 1a and Supplementary Fig. 1a). Tcf1 expression gradually increased in M45-specific T cells (Fig. 1b), concomitant to loss of KLRG1 expression[22] (Supplementary Fig. 1a, b). In M38-specific cells, despite a small increase in Tcf1 expression, the percentage of GFP-expressing cells was lower than in non-inflationary M45-specific cells (Fig. 1b). Since the majority of inflationary CD8 T cells exhibit a $T_{EM}$ phenotype indicated by KLRG1 expression (Supplementary Fig. 1a, b), the low percentage of Tcf1+ cells was not surprising[4]. Also in the spleen, lungs and LN, only a small fraction of both M45- and M38-specific T cells expressed Tcf1 8 days post-infection (Fig. 1c, d), although a slightly higher percentage of Tcf1 expressing cells was found in the LN. However, on day 70 post-infection, the majority of M45-specific T cells expressed Tcf1 in all organs examined (Fig. 1e, f). A small percentage of M38-specific T cells expressed Tcf1 in the spleen and lungs. Strikingly, M38-specific T cells had a higher percentage of GFP+ cells in the LNs (Fig. 1e, f). Comparable percentages of Tcf1+ cells were found in LNs isolated from different anatomical locations (Supplementary Fig. 1c, d), underscoring the LNs as a site where Tcf1 expressing M38-specific CD8 T cells are enriched.

Our findings were confirmed using monoclonal P14 T cell receptor (TCR) transgenic T cells recognizing the LCMV glycoprotein GP33-41 epitope. CD45.1+ P14 Tcf7GFP cells were transferred into CD45.2+ WT mice that were subsequently infected with an MCMV variant expressing the GP33-41 epitope in the ie2 locus (MCMV-ie2-GP33), eliciting a GP33-specific T cell response with inflationary characteristics[29] (Supplementary Fig. 1e). The majority of P14 cells in the LN, but not in the periphery, expressed Tcf1 (Fig. 1g, h, Supplementary Fig. 1f). The GFP signal correlated with Tcf1 protein expression (Fig. 1i). MCMV-ie2-GP33 expresses the viral protein m157 which binds and activates Ly49H receptors on NK cells in C57BL/6 mice. M38-specific responses induced by MCMV-Δm157 were somewhat elevated compared to these responses induced by m157-expressing-MCMV (Supplementary Fig. 1g). However, similar expression of Tcf1 was observed (Supplementary Fig. 1h), indicating that activation of Ly49H+ NK cells did not influence Tcf1 expression. Endogenous GP33-specific T cell responses induced by MCMV-ie2-GP33 accumulated in the blood as well, displaying an inflationary T cell response (Supplementary Fig. 1g). Comparable Tcf1 expression kinetics was found in endogenous and transgenic GP33-specific cells (Supplementary Fig. 1f, h). These results show that Tcf1 is highly expressed in cells with non-inflationary specificity, whereas only in the LN a large fraction of inflationary cells expresses Tcf1.

**Tcf1 expression correlates with a central memory phenotype.** Tcf1 is required for $T_{CM}$ formation[25,26] and Tcf1+ cells share multiple molecular and functional properties with $T_{CM}$ cells[27]. We found that the majority of Tcf1+ cells expressed CD62L and CD127 (Fig. 2a), markers that are expressed by $T_{CM}$ cells, whereas

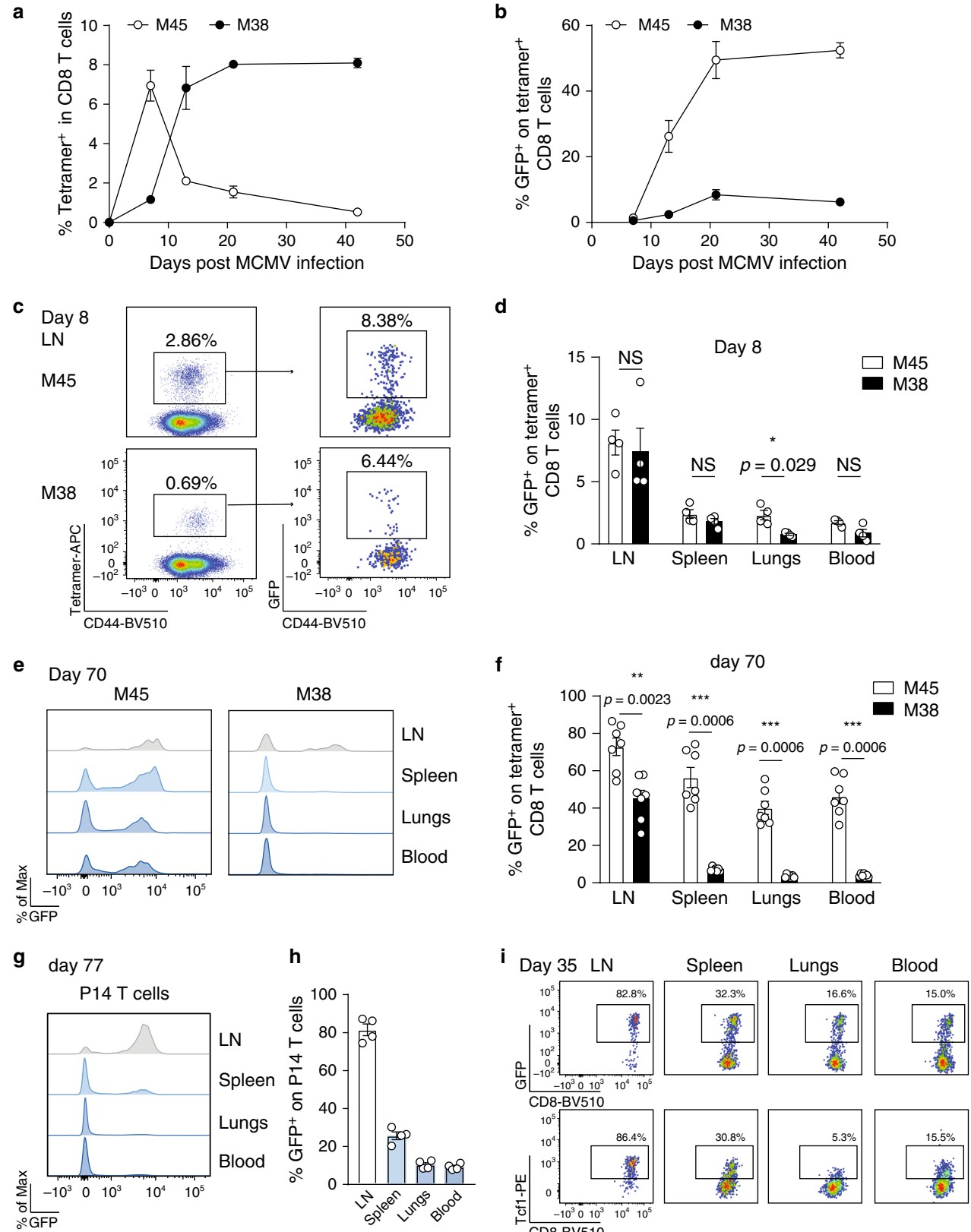

Tcf1⁻ cells expressed the typical $T_{EM}$ markers KLRG1 and CX3CR1 (Fig. 2a). MCMV-specific P14 cells in the LN, mostly expressing Tcf1, were frequently found in the outer periphery of the T cell zone (Fig. 2b), a site where $T_{CM}$ cells are positioned[30]. Some P14 cells were detected in and beneath the B cell follicles (Fig. 2b). A small percentage of MCMV-specific P14 cells

expressed CXCR5, potentially explaining their localization in B cell follicles. CXCR5 was found on both Tcf1⁺ and Tcf1⁻ cells (Supplementary Fig. 2a, b), unlike observations in chronic LCMV infection, where Tcf1 expression was correlated with CXCR5 expression[31]. Both Tcf1⁺ and Tcf1⁻ cells expressed PD-1 in chronic LCMV infection, and Tcf1⁻ cells expressed additional

**Fig. 1 Tcf1 expression in MCMV-specific CD8 T cells.** *Tcf7*[GFP] mice were i.v. infected with $10^6$ PFU MCMVΔm157. **a** Percentages of M45- and M38-specific within CD8 T cells and the frequency of Tcf1 (GFP[+]) cells within tetramer-positive populations (**b**) are shown in the blood as mean ± SEM ($n = 3$-4) of one experiment out of three independent experiments. **c** Cells from the inguinal LN were assed for CD44 and MHC class I tetramer binding 8 days post-infection (left dot plot). Right plots show Tcf1 expression, indicated by GFP, on M45- and M38-specific CD8 T cells. **d** Bar graphs show the percentage of MCMV-specific CD8 T cells expressing Tcf1 in different organs at day 8 post MCMV infection ($n = 4$). **e** Histograms show mean fluorescence intensity of Tcf1 (GFP) on M45- (left plot) and M38-specific (right plot) CD8 T cells at day 70 post-infection. **f** Bar graphs show the percentage of MCMV-specific CD8 T cells expressing Tcf1 in different organs at day 70 post MCMV infection. Pooled data of two experiments are shown ($n = 7$). **g** CD45.1[+] P14 *Tcf7*[GFP] cells were adoptively transferred into naïve WT mice that were subsequently infected with $2 \times 10^5$ PFU MCMV-ie2-GP33. Histograms show mean fluorescence intensity of Tcf1 (GFP) on P14 cells. **h** Percentage of P14 T cells that express GFP are shown at 77 days post-infection ($n = 4$). **i** Flow cytometry plots shows expression of Tcf1 indicated by GFP expression (top row) or by intracellular antibody staining for Tcf1 (bottom row), on P14 T cells 35 days post-infection with $2 \times 10^5$ PFU MCMV-ie2-GP33. All bar graphs represent mean ± SEM ($n = 3$–5 mice) of one experiment out of 2 or 3 independent experiments. Each dot represents an individual mouse. *$p < 0.05$, **$p < 0.01$, ***$p < 0.001$; statistical significance was determined using the two-sided non-parametric Mann–Whitney test. Source data are provided as a Source Data File.

co-inhibitory receptors (e.g. Tim3, Lag3). These markers were hardly found on MCMV-specific CD8 T cells (Supplementary Fig. 2c). Opposed to exhausted T cells in chronic LCMV infection, inflationary Tcf1[−] T cells express low levels of CD27[4] (Supplementary Fig. 2c). The communalities between P14 cells in MCMV and LCMV infection were that CD62L, CD127 and Ly108 expression were restricted to Tcf1[+] cells, whereas KLRG1, CX3CR1 and CD39 were expressed by Tcf1[−] cells. Tcf1[+] cells were better IL-2-producers, with more IL-2-producers in MCMV infection. Tcf1[−] cells produced more IFNγ compared to Tcf1[+] cells in LCMV infection, while both produced comparable levels in MCMV infection. Overall, cytokine production was higher in MCMV-specific CD8 T cells (Supplementary Fig. 2d)

In MCMV infection, splenic Tcf1[+] P14 cells were mostly found in the T cell zone of the white pulp, whereas Tcf1[−] P14 cells localized mainly to the red pulp (Fig. 2c). The relative expression of Tcf1 was inversely correlated to the initial viral inoculum dose, because a higher percentage of M38-specific T cells expressed Tcf1 and exhibited a $T_{CM}$ phenotype in mice infected with a low inoculum dose (Fig. 2d, e), This is in line with previous reports showing that the viral inoculum dose impacts on the extent of memory inflation[32].

**Tcf1 expression is regulated by IL-12 and type I IFNs.** Systemic inflammation mediated via IL-12, suppresses the expression of Tcf1 to facilitate effector cell formation[33]. Both type I IFNs and IL-12 are produced upon MCMV infection[34] and are implicated in effector cell differentiation[35–37]. We addressed the influence of these cytokines on Tcf1 expression by transferring equal numbers of WT, IL12Rβ2[−/−] and IFNAR[−/−] P14 cells (Fig. 3a) into the same hosts that were subsequently infected with MCMV-ie2-GP33. No differences in Tcf1 expression were observed in naïve WT, IL12Rβ2[−/−] and IFNAR[−/−] P14 cells (Fig. 3b). On day 8 post-infection, WT P14 cells were more abundantly found in spleen, lungs and blood, followed by IL12Rβ2[−/−] and IFNAR[−/−] P14 cells (Fig. 3b, c). Strikingly, in the LN, a similar abundance of WT and IL12Rβ2[−/−] P14 cells was found. Moreover, Tcf1 expression was increased in mice that were unable to sense either IL-12 or type I IFNs (Fig. 3d). This effect was most pronounced in LNs, likely due to the abundance of Tcf1 expressing cells at this site. These data show that in MCMV infection, IL-12 and type I IFN contribute to downregulate Tcf1 expression.

**Tcf1[+] cells retain proliferative potential.** The half-life of inflationary T cells is around 10–12 weeks in the periphery[21], however, the inflationary pool is remarkably stable in numbers. To maintain the inflationary pool at high levels, continued supply needs to be guaranteed. We addressed if the Tcf1[+] subset of the inflationary T cell pool fulfils this function by seeding the pool of peripheral effector-like T cells. Tcf1[+] and Tcf1[−] P14 cells were sorted from MCMV-ie2-GP33-infected mice based on GFP expression. Equal numbers of P14 cells were separately transferred into infection-matched recipients (Fig. 4a). To track proliferation history, the transferred cells were labelled with a cell proliferation dye (CPD). After 4 weeks, we assessed the pool of T cells derived from the transferred Tcf1[+] and Tcf1[−] P14 subsets. Tcf1[+] cells divided more frequently than Tcf1[−] cells (Fig. 4b, c), including all cells that divided at least once. Strikingly, Tcf1[+] cells gave rise to a pool of cells after several cell divisions that lost Tcf1 and gained KLRG1 expression (Fig. 4d, e). Also in naïve mice, Tcf1[+] cells divided more frequently than Tcf1[−] cells (Fig. 4b). However, no Tcf1[−] cells were detected after transfer of Tcf1[+] cells in naïve hosts, indicating that the generation of the Tcf1[−] population was driven by antigen and the proliferation in naïve recipients reflected homeostatic proliferation (Fig. 4b, d).

To confirm that the production of Tcf1[−] cells from Tcf1[+] cells also applied to CD8 T cells with specificity for a natural MCMV epitope, we crossed *Tcf7*[GFP] mice with the TCR transgenic Maxi mouse, in which all CD8 T cells express a transgenic TCR recognizing the MCMV M38[316-323] epitope[8]. Following adoptive transfer and MCMV infection, only a small fraction of Maxi cells in the periphery expressed Tcf1, whereas a larger proportion expressed Tcf1 in the LN (Supplementary Fig. 3a). The percentage of Maxi cells that expressed Tcf1 was lower compared to P14 cells in the LN upon MCMV-ie2-GP33 infection (Fig. 1g and Supplementary Fig. 3a), probably owing to more frequent/abundant expression of the M38[316-323] epitope upon viral reactivation than the GP[33-41] epitope. Tcf1[+] and Tcf1[−] Maxi cells were isolated from MCMV infected mice, labelled with CPD and transferred into infection-matched recipients (Supplementary Fig. 3b). Tcf1[+] Maxi cells divided more frequently as Tcf1[−] cells in both naïve and MCMV infected mice (Supplementary Fig. 3c, d). However, in MCMV infected hosts, around 75% of the transferred Tcf1[+] Maxi cells had divided at least once compared to 40% in naïve mice (Supplementary Fig. 3c, d). In MCMV infected hosts, Tcf1[+] cells experienced more cell divisions and gave rise to a pool of Maxi cells that out-diluted the CPD and expressed KLRG1 (Supplementary Fig. 3e, f). These results show that, upon antigen encounter, Tcf1[+] cells proliferate extensively, and feed into the pool of inflationary T cells that express KLRG1 and lose Tcf1 expression.

Upon transfer of Tcf1[−] P14 or Maxi cells, hardly any cells were detected in the LNs (Fig. 4f and Supplementary Fig. 3g), indicating that Tcf1[−] cells are hampered in LN homing. It is currently unclear where viral reactivation events are sensed. Tcf1[+] cells respond to antigen encounter by proliferation and the LN is a site where these Tcf1[+] cells are enriched, making it likely that viral reactivation events are sensed here. As Tcf1[−] cells are

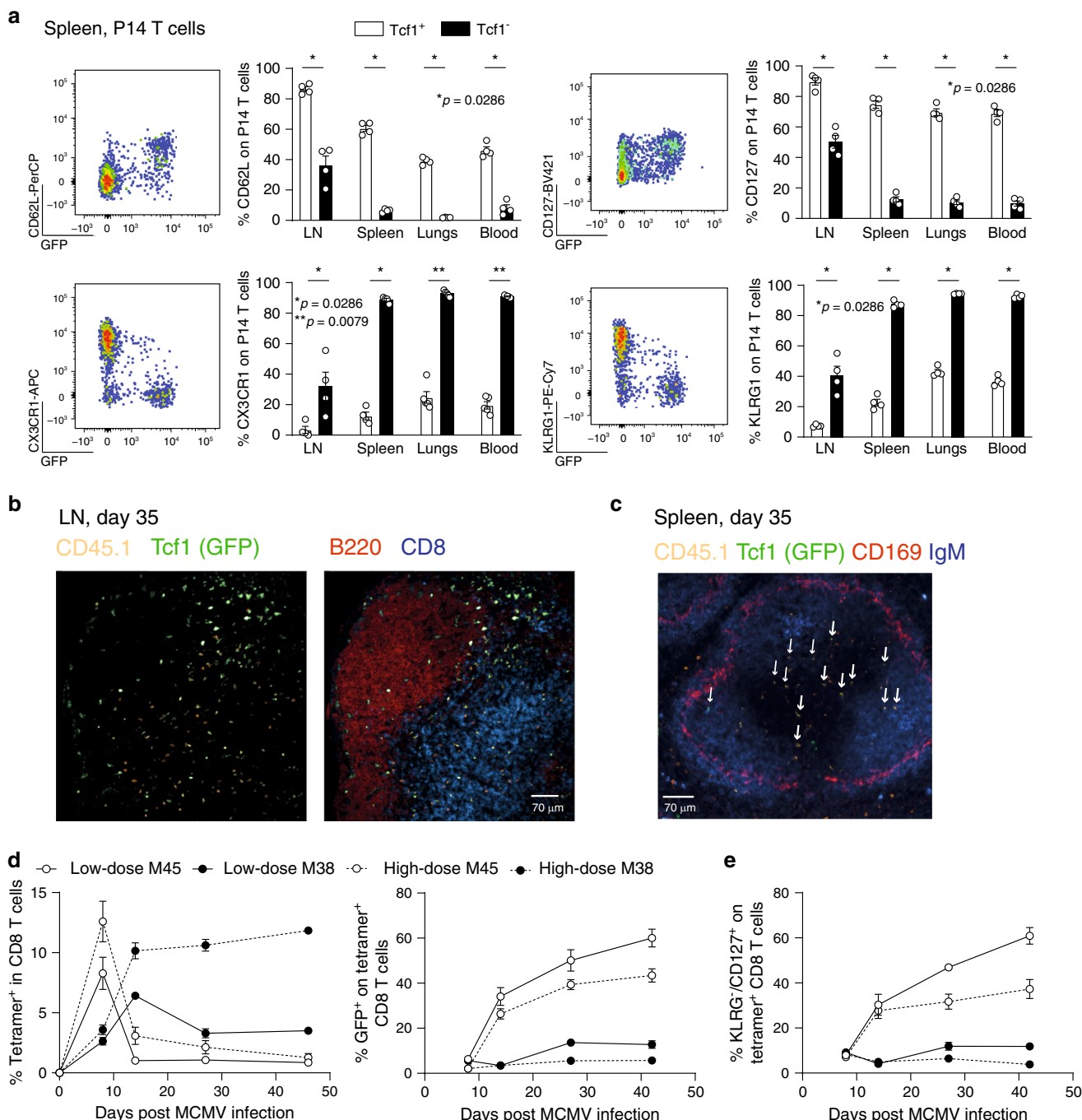

**Fig. 2 Tcf1+ cells express markers associated with a central memory phenotype. a** CD45.1+ *Tcf7*GFP P14 cells were adoptively transferred into WT mice that were subsequently infected with 2 × 10^5 PFU MCMV-*ie2*-GP33. Representative flow cytometry plots show the expression of GFP (Tcf1) and the indicated cell surface markers on P14 T cells in the spleen in latent MCMV infection. Bar graphs show the percentage of Tcf1+ (GFP+) and Tcf1− (GFP−) P14 cells expressing the indicated cell surface markers. All bar graphs represent mean ± SEM and each dot represents an individual mouse (*n* = 4–5). **b** Immunofluorescence staining of an inguinal LN section is shown 5 weeks post MCMV-*ie2*-GP33 infection with CD45.1 (P14 T cells) (orange), GFP representing Tcf1 (green), B220 (red) and CD8 T cells (blue). Scale bar is 70 μm. Section is representative of three individual mice. **c** Immunofluorescence staining of a splenic section is shown 5 weeks post MCMV-*ie2*-GP33 infection with CD45.1 (P14 T cells) (orange), GFP representing Tcf1 (green), CD169 (red) and IgM (blue). Scale bar is 70 μm. Arrows indicate Tcf1+ cells determined by GFP expression. Section is representative of three individual mice. **d** *Tcf7*GFP mice were infected with a low (10^3 PFU) or a high (10^6 PFU) dose of MCMVΔm157. The percentage of M45- and M38-specific CD8 T cells, determined by MHC Class I tetramer binding (left graph) and the percentage of tetramer+ CD8 T cells expressing GFP (right graph) is shown in the blood as mean ± SEM (*n* = 4). **e** The percentage of tetramer+ CD8 T cells expressing CD127 in the blood is shown as mean ± SEM (*n* = 4). **a–e** One experiment out of 2 or 3 independent experiments is shown (*n* = 3–5 mice). **p* < 0.05, ***p* < 0.01; statistical significance was determined using the two-sided non-parametric Mann–Whitney test. Source data are provided as a Source Data File.

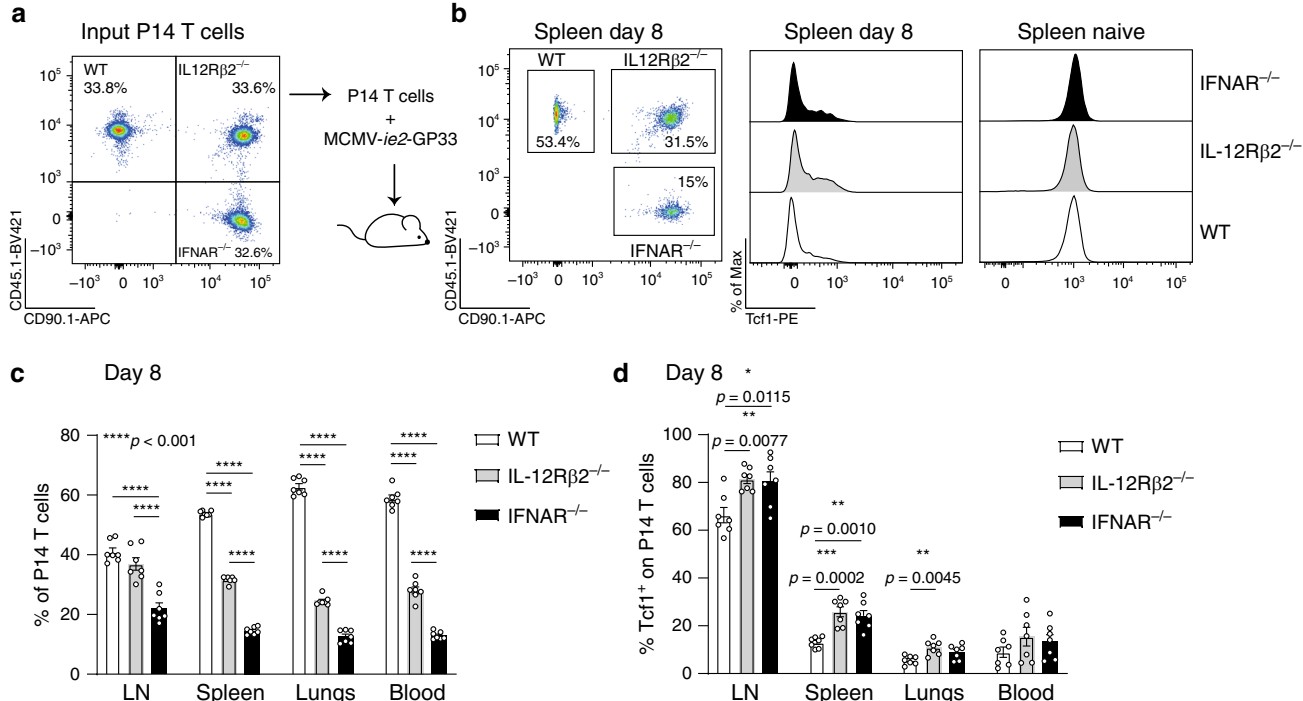

**Fig. 3 Type I IFNs and IL12 regulate Tcf1 expression in MCMV-specific CD8 T cells. a** Experimental setup: equal numbers of CD45.1$^+$ WT, CD45.1$^+$ CD90.1$^+$ IL12Rβ2$^{-/-}$ and CD90.1$^+$ IFNAR$^{-/-}$ P14 T cells, in total 10$^4$ cells, were adoptively transferred into the same hosts that were subsequently infected with 2 × 10$^5$ PFU MCMV-*ie2*-GP33. Representative flow cytometric plot shows the portion of WT, IL12Rβ2$^{-/-}$ and IFNAR$^{-/-}$ P14 T cells in the mix that was initially injected into the mice. **b** Representative flow cytometric plot shows the portion of WT, IL12Rβ2$^{-/-}$ and IFNAR$^{-/-}$ P14 T cells in the spleen 8 days post-infection. Histogram shows the expression of Tcf1 (by intracellular staining) in the distinct P14 T cells. **c** The percentage of WT, IL12Rβ2$^{-/-}$ and IFNAR$^{-/-}$ P14 T cells in the total transgenic T cell pool is shown (n = 7). **d** Percentage of P14 T cells that expresses Tcf1 is shown (n = 7). All bar graphs represent mean ± SEM and each dot represents an individual mouse shown from pooled data of two independent experiments (n = 3–4 mice). *p < 0.05, **p < 0.01, ***p < 0.001 ****p < 0.0001; statistical analyses were performed using one-way ANOVA followed by Tukey's multiple comparisons test. Source data are provided as a Source Data File.

barely found within the LNs, this could explain their diminished proliferation potential. We examined the proliferative capacity of Tcf1$^+$ Maxi cells upon abrogation of LN homing using αCD62L-blocking antibodies. Fewer Maxi cells were found in the LNs (Supplementary Fig. 3g). However, the proliferative capacity of Tcf1$^+$ cells was not affected (Supplementary Fig. 3h). These results indicate that the potent proliferative capacity of Tcf1$^+$ cells is not due to their enhanced ability to enter the LNs.

Upon CD62L-blocking, the entry of Tcf1$^+$ cells in the spleen was not affected. This could explain why the proliferation capacity of Tcf1$^+$ cells was unaltered when LN entry was blocked. We determined whether memory inflation would be affected when also recirculation through the spleen would be abrogated. One approach to interfere with lymphocyte exit from secondary lymphoid organs is to use the sphingosine 1-phosphate (S1P) analogue FTY720, which inhibits S1P-S1PR-axis-mediated lymphocyte egress. However, upon CMV infection FTY720 only sequestered MCMV-specific CD8 T cells in the LNs and not in the spleen (Supplementary Fig. 4a). To assess whether blocking of homing of MCMV-specific CD8 T cells to T cell zones of secondary lymphoid organs, where the majority of Tcf1$^+$ cells are found and possibly reactivation of Tcf1$^+$ cells could occur, would have an impact on memory inflation, we adoptively transferred WT and CCR7$^{-/-}$ Mini cells. Mini cells are TCR Vβ-chain transgenic cells in which approximately 10% is specific for the MCMV M38$_{313-323}$ epitope determined by M38-tetramer staining, due to pairing with endogenous TCR Vα-chains[8]. CCR7 deficiency leads to hampered migration towards CCL19 and CCL21-rich T cell zones in LNs and spleen. During acute MCMV

infection, expansion of CCR7$^{-/-}$ Mini cells was about 3-fold reduced compared to WT cells. This difference markedly increased during MCMV latency, eventually reaching a 23-fold difference (Supplementary Fig. 4b, c). Also in the lungs, spleen and LNs, the number of CCR7$^{-/-}$ Mini cells was diminished (Supplementary Fig. 4d). A higher percentage of WT Mini cells had a more terminally differentiated phenotype (Supplementary Fig. 4e). These results indicate that CCR7 expression on inflationary T cells promotes their inflation, both numerically and phenotypically.

**Tcf1$^+$ cells maintain the inflationary T cell pool**. We next addressed whether Tcf1$^+$ cells are necessary for the maintenance of the inflationary T cell pool. *Tcf7*$^{DTR-GFP}$ P14 cells were used, where all Tcf1$^+$ cells express the diphtheria toxin receptor (DTR) and GFP. By administration of Diphtheria toxin (DT), DTR-expressing cells are selectively depleted[38]. CD45.2$^+$ *Tcf7*$^{GFP-DTR}$ P14 cells were transferred into CD45.1$^+$ mice that were infected with MCMV-*ie2*-GP33. At day 8 post-infection, four consecutive shots of DT were administrated (Fig. 5a). This resulted in selective depletion of Tcf1$^+$ P14 cells, indicated by absence of GFP$^+$ cells (Fig. 5b) and Tcf1 protein expressing cells (Fig. 5b). Importantly, one day after administration of the first DT dose (representing day 9 post-infection), no differences were found in the number of GFP$^-$ P14 cells, indicating that DT administration did not impact these cells directly (Fig. 5c, d). We followed the kinetics of the GFP$^-$ cells without including the GFP$^+$ P14 cells as this would reflect DT-administration-induced differences in the population size of GFP$^+$ P14 cells. Longitudinal tracking of

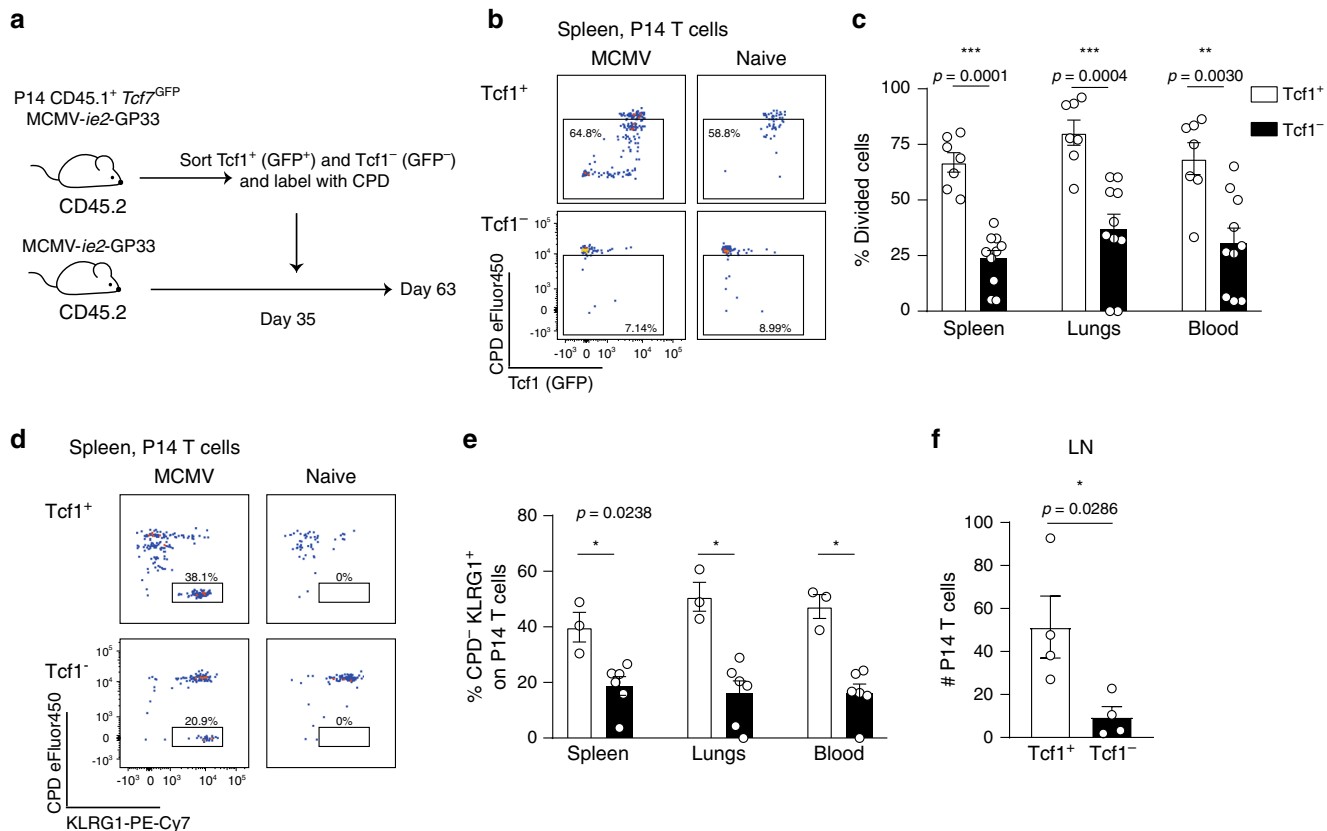

**Fig. 4 Tcf1+ cells have proliferative capacity and give rise to Tcf1− KLRG1+ cells. a** Experimental setup: *Tcf7*GFP P14 T cells were adoptively transferred into CD45.2+ mice that were subsequently i.v. infected with 2 × 10^5 PFU MCMV-*ie2*-GP33. After 5 weeks, Tcf1+ and Tcf1− P14 cells were sorted based on GFP, subsequently labelled with a cell proliferation dye and 5 × 10^4 P14 T cells were adoptively transferred into infection-matched recipients. Naïve hosts received 1 × 10^5 P14 T cells. Four weeks post adoptive transfer, the proliferation profile of the P14 T cells was determined. **b** Representative flow cytometry plot shows the proliferation history based on CPD dilution versus Tcf1 expression indicated by GFP on P14 T cells in the spleen. **c** The percentage of P14 cells in the spleen, blood and lungs that have proliferated at least once based on their CPD dilution, 28 days after adoptive transfer, is shown. Pooled data from two experiments are shown (n = 7–10). **d** Representative flow cytometry plot shows the proliferation history based on CPD dilution versus KLRG1 expression on P14 T cells in the spleen (n = 3–6). **e** The percentage of P14 cells that has out-diluted the CPD and expresses KLRG1 is shown. **f** The total number of P14 T cells in the LN is shown (n = 4). All bar graphs represent mean ± SEM, each dot represents an individual mouse. Data are representative from three independent experiments (n = 3–6 mice). *p < 0.05, **p < 0.01, ***p < 0.001; statistical significance was determined using the two-sided non-parametric Mann–Whitney test. Source data are provided as a Source Data File. .

GFP- (Tcf1−) P14 cells in the blood revealed an accumulation of these cells in mice without DT treatment. Strikingly, in Tcf1+ depleted mice this accumulation was not observed (Fig. 5e). Also, in the spleen and lungs a diminished amount of Tcf1− cells was found 100 days post-infection when mice were devoid of Tcf1+ P14 cells (Fig. 5f). In mice without DT treatment, GFP+ cells were detected in the LN, whereas in DT treated mice these cells were hardly observed (Fig. 5g). Although the depletion of Tcf1+ cells early after DT administration was highly efficient, 100 days later, a small population of Tcf1+ cells was detected, potentially explaining the residual maintenance of the P14 population at low levels (Fig. 5g). These data support the notion that Tcf1+ cells give rise to the peripheral pool of Tcf1− effector cells and upon depletion of the Tcf1+ population, memory CD8 T cell inflation is hampered.

**The Tcf1+ pool has a larger clonal diversity**. Our results indicate that within the inflationary T cell pool, Tcf1− cells are derived from the Tcf1+ population. This suggests clonal overlap between the TCR usage of Tcf1+ and Tcf1− cells. We applied next generation sequencing to delineate the TCR-Vβ chain repertoire of sorted Tcf1+ and Tcf1− M38-specific cells from spleen, lungs,

mediastinal LN and a pool of LNs (Fig. 6a). Strikingly, the Tcf1+ pool contained more unique clones than the Tcf1− population, defined by unique CDR3 amino acid sequence (Supplementary Fig. 5a). This difference was evenmore pronounced when we normalized to the total number of isolated cells (Fig. 6b). For comparison, the number of unique clonotypes in the blood, reflecting the total T cell repertoire, was ~100 times higher than the number of clones within the M38-specific population (Supplementary Fig. 5b). The larger clonal diversity of the Tcf1+ population was reflected in all tissues (Fig. 6c). Interestingly, the highest diversity was found in the LNs (Fig. 6c). The total response was polyclonal, as many different TCR V-β-genes were recruited in the response in both the Tcf1+ and Tcf1− population. TRBV1 was the most utilized V-gene across unique clones (Fig. 6d), followed by TRBV4 and TRBV5 which is consistent with other reports that found these genes in M38-specific responses[8,39]. The overall TCR V-β-gene usage did not differ drastically between the Tcf1+ and Tcf1− population (Fig. 6d). Despite the comparable selection of the TCR V-β-gene usage, more unique clones per TCR V-β-gene were found in the Tcf1+ population (Fig. 6e), reflecting a larger diversity in the TCR repertoire. Also for the CDR3 length selection, more unique

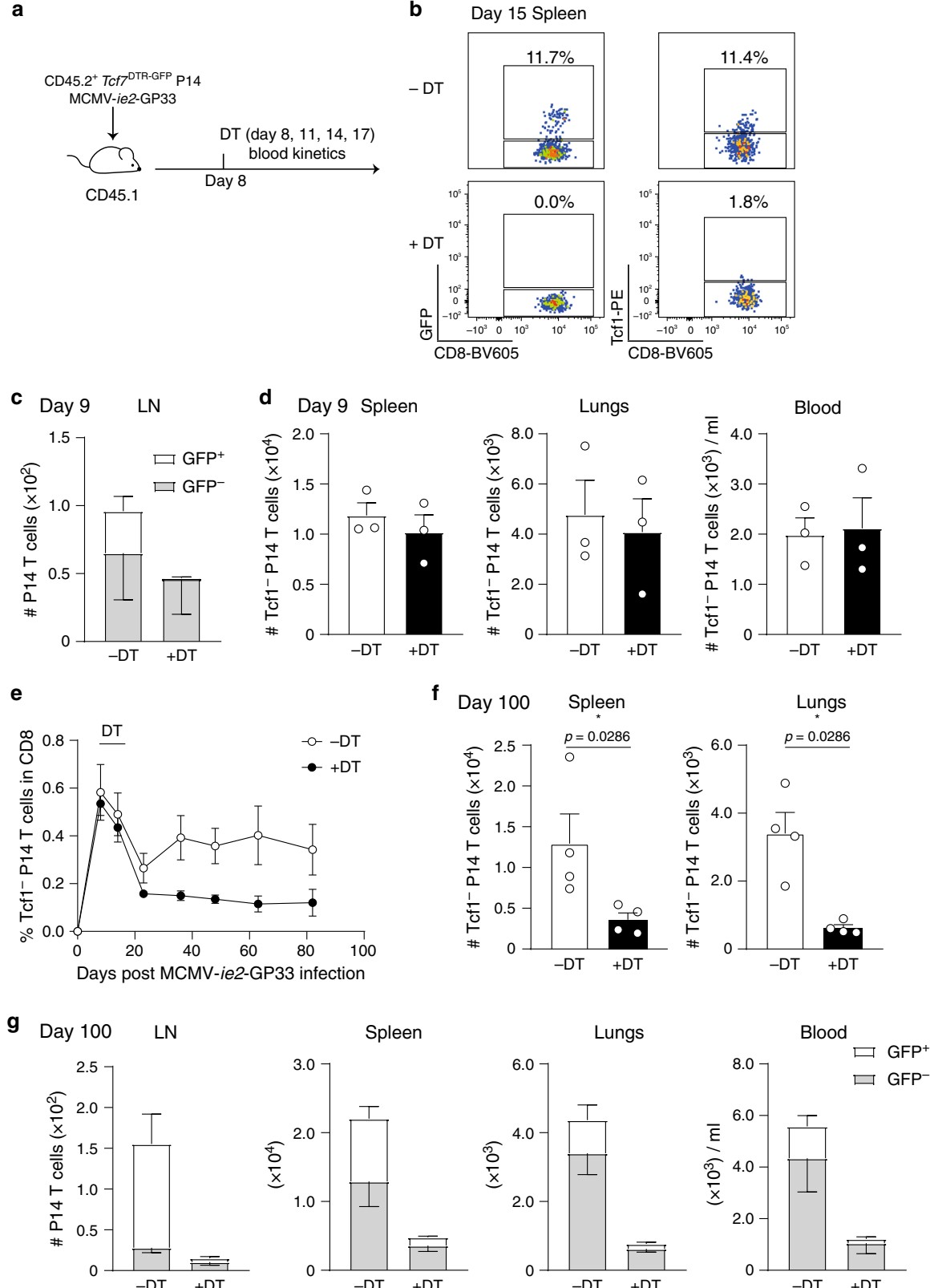

clones were found in the Tcf1$^+$ population (Supplementary Fig. 5c). The distribution amongst the CDR3 length selection did not follow a gaussian distribution as in naïve mice[40], but was skewed towards a CDR3 length of 17 amino acids (Supplementary Fig. 5d, e). This was observed in the total pool of unique M38-specific Tcf1$^+$ and Tcf1$^-$ sequences and in individual tissues (Supplementary Fig. 5d, e). Unique clonotypes isolated from the blood, representing the total T cell repertoire, gave the expected gaussian distribution for the CDR3 length selection (Supplementary Fig. 5f)[40]. These results show that the M38-specific Tcf1$^+$ pool has a larger clonal diversity than the Tcf1$^-$ population.

**Fig. 5 Tcf1+ cells are critical for maintaining the inflationary T cell pool at high numbers. a** Experimental setup: $4 \times 10^3$ CD45.2+ $Tcf7^{DTR-GFP}$ P14 T cells were adoptively transferred in CD45.1+ hosts that were subsequently infected with $2 \times 10^5$ PFU MCMV-*ie2*-GP33. On day 8, 11, 14 (and 17) post-infection one group of mice received 1 µg DT via an i.p. injection. **b** Representative flow cytometry plot shows Tcf1 expression indicated by GFP (left) and determined by intracellular protein staining (right) on CD45.2+ P14 T cells in the spleen, 15 days post-infection in mice that were left untreated (upper row) or in mice that received DT (bottom row). **c** The total number of GFP+ and GFP− P14 cells in the inguinal LNs was determined 9 days post-infection (1 day after the first administration of DT) in mice with and without DT treatment. Bar graphs represent mean + SEM (GFP+) and mean - SEM (GFP−) (n = 3). **d** The total number of GFP− P14 cells was determined 9 days post-infection (1 day after the first administration of DT) in mice with and without DT treatment. Bar graphs represent mean + SEM (n = 3). **e** The percentage of Tcf1− P14 T cells (GFP−) in the CD8 T cell pool is shown in the blood in mice with or without DT treatment (n = 4). **f** The total number of Tcf1− P14 cells, indicated by GFP expression, is shown in the lungs and in the spleen 100 days post-infection as mean + SEM (n = 4). **g** The total number of Tcf1+ (GFP+) and Tcf1− (GFP−) P14 cells, is shown in the inguinal LNs, spleen, lungs and blood 100 days post MCMV-*ie2*-GP33 infection. Bar graphs represent mean + SEM (GFP+) and mean - SEM (GFP−) (n = 4). Data are shown from one representative experiment out of two (**b**–**d**) or out of 4 (**e**, **f**) independent experiments (n = 3–9 mice). *$p < 0.05$; statistical significance was determined using the two-sided non-parametric Mann–hitney test. Source data are provided as a Source Data File.

**Abundant Tcf1− clones are reflected in the Tcf1+ pool.** We compared clonal overlap between the M38-specific Tcf1+ and Tcf1− population focussing on expanded clones found across tissues above a threshold of 0.5% of the repertoire based on clonal frequency. Around 50% of the expanded unique Tcf1− clones were found in the Tcf1+ pool (Fig. 7a, b) within individual mice. When comparing the Tcf1− population of one mouse with the Tcf1+ population of another mouse, the overlap was around 20% (Fig. 7a, b) (quantifications were made across all combinations of mice). For comparison, the overlap between the total blood compartment among mice was only 5%. Similar results were obtained when taking those clones with clonal frequencies greater than 1% of the repertoire or when all unique clones regardless of clonal frequency were included (Supplementary Fig. 6a, b). Further analysis revealed a strong correlation between clonal frequencies of all overlapping clones, highlighting that if a clone was highly abundant in the Tcf+ compartment it was similarly abundant in the Tcf1− population (Fig. 7c). These results demonstrate a mouse-specific clonal overlap between the M38-specific Tcf1+ and Tcf1− populations.

To understand the stability of individual clones over time, we quantified how abundant M38-specific clones were observed in the blood on several timepoints post-infection (Fig. 6a). We selected the 20 most expanded M38-specific clones found across pooled organs in the Tcf1+ and Tcf1− population in each mouse at day 120 post-infection and determined the frequency of detection in the blood. The majority of the top 20 Tcf1− clones was found at all three timepoints in the blood, whereas half of the Tcf1+ clones was not detected at any timepoint (Fig. 7d), indicating that these cells are circulating less or are present in the circulation at a lower frequency making detection more difficult. The clones that were found across all timepoints were remarkably stable in frequency and included some overlapping clones that were found in both the Tcf1+ and Tcf1− compartment (Fig. 7e), indicating that within the anaysed time period there is little clonal diversification within the pool of expanded M38-specific cells.

These data show significant clonal overlap between the Tcf1+ and Tcf1− population, that the most expanded clones are found in comparable abundance in both populations, and that within a time period of 60 days there is little variation in the clonal composition of circulating M38-specific CD8 T cells.

## Discussion

Memory inflation refers to the accumulation of large numbers of functional effector-like cells in peripheral tissues and blood. Maintaining the inflationary T cell pool is a dynamic process, with continuous recruitment of new cells replenishing the peripheral pool of effector-like cells. Naïve cells hardly contribute to this refuelling[22]. Likewise, thymectomized mice do not show impaired memory inflation[41]. We found that a small subset of inflationary T cells expresses Tcf1, is proliferation competent and gives rise to peripheral Tcf1− cells. Recently, we showed that the number of early primed KLRG1− CD8 T cells has an impact on the degree of memory inflation[20]. The few Tcf1+ cells detected early in MCMV infection could represent the same subset.

Similar to other studies[33], we found that Tcf1 expression is influenced by pro-inflammatory cytokines upon effector cell differentiation. Redundant roles are described for IL-12 and type I IFNs[42]. Therefore, it would be interesting to examine the effects on Tcf1 expression upon a combined deficiency for IL-12- and type I IFN-mediated-signalling. Whether IL-12 and/or type I IFNs are required when memory Tcf1+ cells re-encounter antigen and produce Tcf1− effector-like cells, remains to be investigated. It is clear however that several cell divisions are necessary before Tcf1 expression is suppressed[33,43].

A larger diversity in the TCR repertoire of M38-specific cells is found in the LNs. If MCMV reactivation events are sensed at this site is unclear, as both LNs and the circulation are proposed[8,44]. Latent viral genomes are found in CD31+ cells in the lungs[45], and the majority of inflationary cells undergoing antigen-driven division are in the vasculature[44]. Besides the LN, we observed divided Tcf1+ cells in the blood, lungs and spleen, and the percentage of dividing cells was unaltered upon abrogation of LN homing. However, presence of proliferating cells in the vasculature does not prove that antigen recognition occurs here, as T cells activated in secondary lymphoid organs will exit within a few days and hence be present in the vasculature. Our data indicate that access to LNs is not necessary for memory inflation. In this situation reactivation of Tcf1+ cells in the spleen might generate Tcf1− cells. FTY720 blocks T cell egress from secondary lymphoid organs, however, FTY720 treatment only had a minimal impact on T cell inflation[44]. The observation period in this study was rather short (5 weeks), compared to a half-life of 10-12 weeks for peripheral inflationary T cells[21]. Furthermore, FTY720 treatment only prevents egress of activated MCMV-specific CD8 T cells from LNs and not from the spleen (Supplementary Fig. 4a). We found that CCR7-deficient T cells poorly contributed to inflation, presumably because these cells are unable to migrate towards CCL19- and CCL21-rich T cell zones of secondary lymphoid organs (also the site were Tcf1+ cells are found), where they either receive survival signals or antigen re-encounter might take place. The survival of peripheral inflationary Tcf1− cells depends on IL-15 trans-presentation, and not on IL-2- or IL-7-mediated signals[21]. As Tcf1+ cells co-express CD127, it is possible that in secondary lymphoid organs these cells depend on IL-7-mediated signals provided by the FRCs for their survival[46,47].

A model for memory inflation was proposed in which cells exist that do not express LN homing molecules but retain proliferative potential[48] and express intermediate levels of CX3CR1. We hardly detected cells with an intermediate CX3CR1 expression, but this

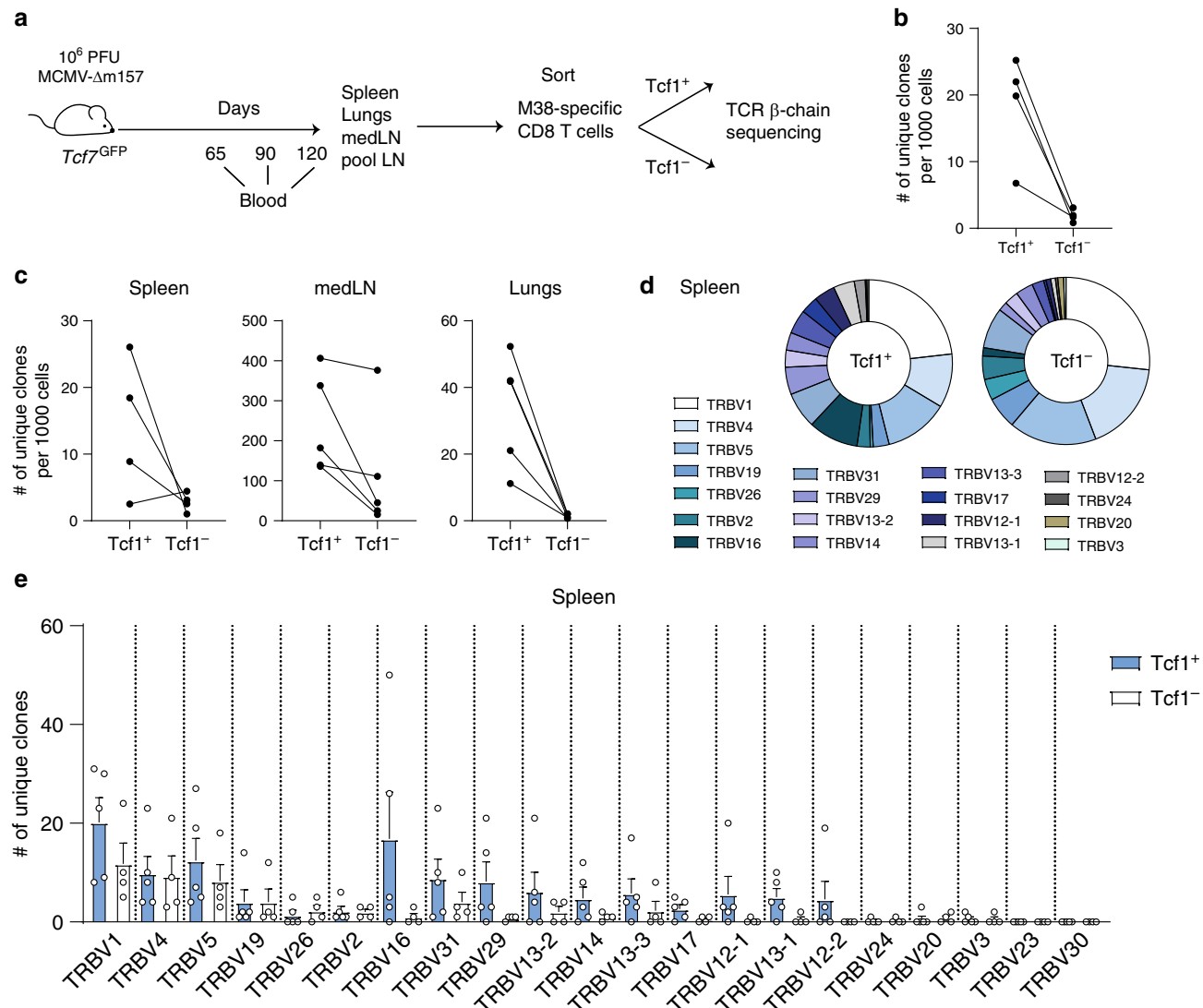

**Fig. 6 M38-specific Tcf1+ cells have a larger clonal diversity than the Tcf1− cells. a** Experimental setup: *Tcf7*GFP mice were i.v. infected with 10[6] PFU MCMV-Δm157. At indicated times blood was collected. At the terminal timepoint, Tcf1+ and Tcf1− cells were sorted from the M38-specific population isolated from spleen, lungs and LNs, and the TCR-V-β chain repertoire was determined by next generation sequencing. **b** The number of unique clones, determined by unique amino acid sequence of the CDR3, per 1000 cells is shown for the Tcf1+ and the Tcf1− M38-specific population. Each line indicates an individual mouse and lines connect data from the same mouse (*n* = 4). **c** The number of unique clones per 1000 cells is shown for the Tcf1+ and the Tcf1− M38-specific population in spleen, mediastinal LN and the lung. Each line indicates an individual mouse and lines connect data from the same mouse (*n* = 4–5). **d** Pie charts indicate the composition of the M38-specific population, where each colour represents the percentage of clones that uses a specific TCR V-β-gene in the spleen at day 120 post-infection. Data show average percentage with *n* = 5 mice (Tcf1+) and *n* = 4 (Tcf1−). **e** Bar graphs represents the total number of unique clones, determined by unique amino acid sequence of the CDR3, that is found within each TCR V-β-gene in the spleen at day 120 post-infection. Data represent mean + SEM with *n* = 4 or 5 mice per group from one experiment. Source data are provided as a Source Data File.

population was characterized using a CX3CR1-reporter mouse that greatly facilitates its detection[49]. Our experiments clearly indicate that Tcf1+ cells are superior in proliferation than Tcf1− cells, although we noticed some proliferation of Tcf1− P14 cells, but not of Tcf1− Maxi cells. The Tcf1+ population in LNs is relatively smaller in Maxi cells than in P14 cells. This might be explained by differences in antigen abundance of the natural CMV M38[316-323]-epitope and the introduced GP[33-41]-epitope. A low MCMV infection dose resulted in a higher percentage of Tcf1+ cells within the virus-specific T cell populations. This implies that the viral load correlates with the percentage of Tcf1+ cells. More viral reactivation events will likely result in faster conversion from Tcf1+ into Tcf1− cells. The cellular source that presents viral antigen upon viral reactivation is unknown, but is of non-hematopoietic origin[8,9], excluding that antigen derived from viral reactivation events in the periphery is transported by hematopoietic cells to the LN.

We observed clonal overlap between Tcf1+ and Tcf1− cells, with a larger TCR repertoire diversity in the Tcf1+ population. Clonal TCR analysis of HCMV-specific CD8 T cells did not show much overlap between T[CM] (IL7Rα+) and T[EM] cells (IL7Rα−) in the LNs and blood, as many clones were only found in one compartment[50]. The spleen was not included in this study and therefore clones might have been underrepresented. There is quite some heterogeneity in the frequency and tissue distribution of HCMV-specific CD8 T cells amongst individuals[51] making it necessary to include multiple tissues to get the full spectrum of clonotypes. If a subset of HCMV-specific inflationary CD8 T cells

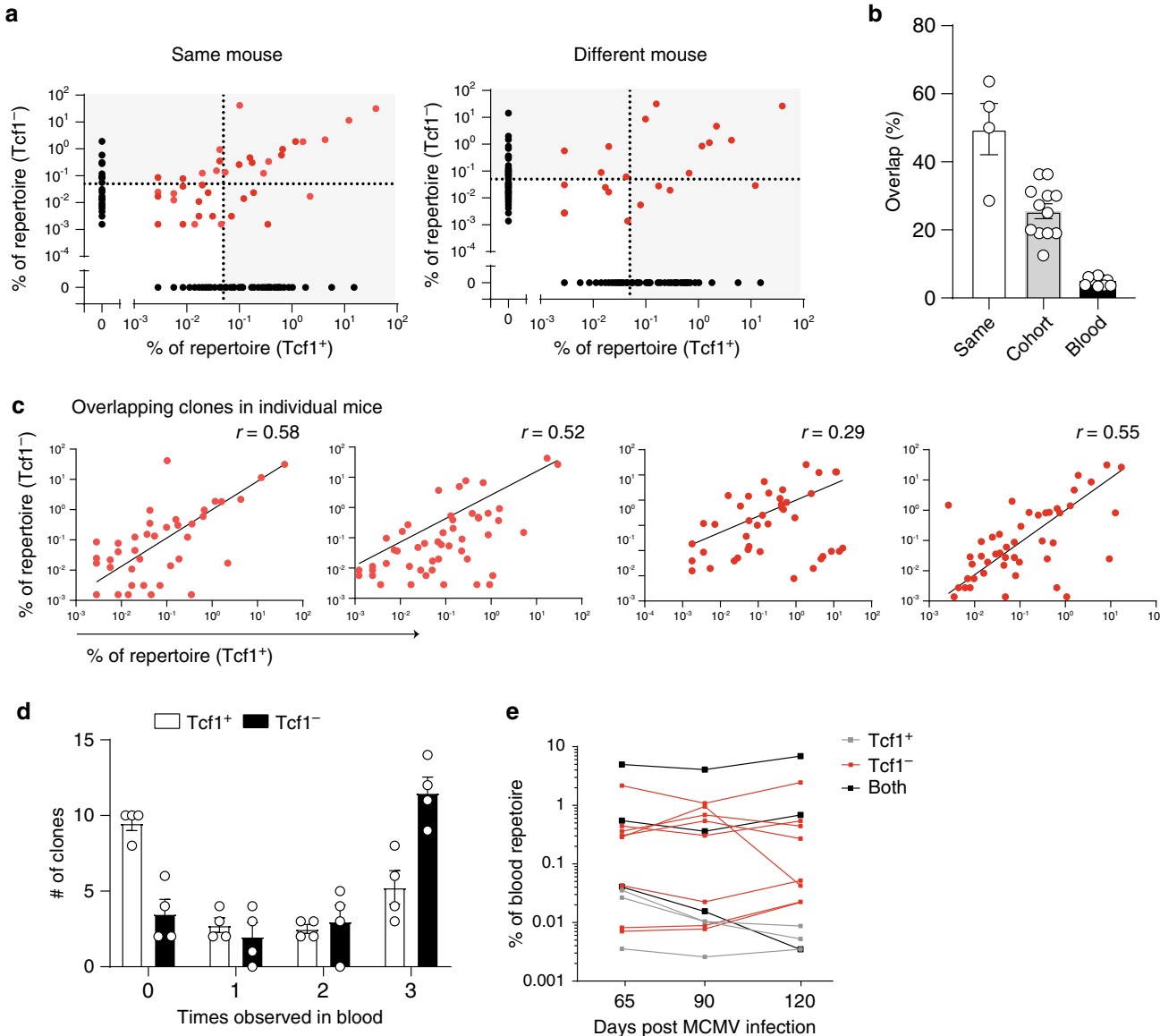

**Fig. 7 Abundant Tcf1+ clones are found in the Tcf1− population. a** Plot indicates all unique clones, represented by a dot, in the M38-specific Tcf1+ and the Tcf1− population of the pooled repertoire (combined spleen, lungs and LNs). Black dots are only found in one compartment. Red dots indicate overlapping clones. Dotted line indicates the 0.5% threshold. Left plot shows the overlap of the Tcf1+ clones and the Tcf1− clones within the same mouse. Right plot shows the overlap between the Tcf1+ from one mouse and Tcf1− clones from a different mouse. **b** Bar graphs shows the percentage of Tcf1− clones that is found within the Tcf1+ compartment compared to the same mouse or compared to a different mouse. Clones that were found above the threshold of 0.5% were included, indicated in the highlighted squares in A. Blood indicates the percentage of overlap of all unique clones between different mice at day 90 post-infection (n = 4). **c** Plots show all overlapping clones found in both the M38-specific Tcf1+ and the Tcf1− population. Each dot represents a unique clone. Each plot represents one mouse, Spearman correlation is indicated. **d** The top 20 most expanded clones from the pooled organs were selected in both the M38-specific Tcf1+ and the Tcf1− population. Bar graphs shows how often these clones were detected across the three blood timepoints (n = 4). **e** Representative plot of one mouse shows the clones that were found across all timepoints in the blood, as the frequency within the blood repertoire in time. Clones that were found in the top 20 of both the Tcf1+ and the Tcf1− population are indicated in black. All bar graphs represent mean ± SEM, n = 4 mice per group. Source data are provided as a Source Data File.

expresses Tcf1 remains to be investigated, but is likely based on the following observations. HCMV-specific CD8 T cells isolated from LNs have a higher proliferative capacity than similar cells obtained from the blood[52]. Increased Tcf1 expression is found in memory CD8 T cells from human LNs compared to other tissues and these memory cells have a higher clonal diversity[52]. Furthermore, Tcf1 marks human memory CD8 T cells capable of self-renewing and upon proliferation these Tcf1+ cells give rise to Tcf1− cells[53,54].

Abundant Tcf1+ clones in the M38-specific response were reflected in the Tcf1−pool. It is conceivable that the higher the abundance of a Tcf1+ clone, the more chance it has to encounter antigen, leading to proliferation of that specific clone and Tcf1− progeny. Despite large overlap, we did not detect for each Tcf1− a Tcf1+ clone. Our TCR repertoire analysis did not include all LNs or the bone marrow, also a site where memory T cells[55] and M38-specific Tcf1+ cells are found (own observation). M38-specific cells expressing intermediate levels of GFP were also excluded,

potentially omitting some clones. We recently showed that high avidity T cells are initially recruited into the inflationary pool[20]. Low avidity HCMV-specific CD8 T cells are increased in older individuals[56], suggesting recruitment of these cells at a later stage, likely when the pool of high avidity cells is depleted. Determining the avidity of endogenous M38-specific Tcf1[−] clones in time would be interesting, however, the frequency of specific clones in the blood was quite stable, admittedly restricted to a relatively short observation period. Whether the time period of analysis could be expanded in mice to an extent that allows to observe substantial changes in clonal composition remains open, as in humans there is clonal stability of HCMV-specific CD8 T cells over a period of at least 5 years[57].

High abundance of antigen, a characteristic of chronic infections and cancers, promotes T cell exhaustion. Tcf1[+] cells maintain the ongoing exhausted CD8 T cell response in chronic LCMV infection in mice and in Hepatitis C virus infected individuals[27,54,58]. Furthermore, Tcf1[+] cells localize to tumours, and respond upon checkpoint blockade immunotherapy by proliferation[38,59]. Although exhaustion and memory inflation are two distinct outcomes of a T cell response[60], also in CMV infection, considered as a low level persistent virus, Tcf1[+] cells are critical in maintaining the ongoing CD8 T cell response.

In animal models, CMV-based vaccines encoding foreign antigens were used for immunization against infections and for cancer immunotherapy[61]. CD8 T cell responses towards these introduced epitopes were induced and their protective capacity was directly correlated to the size of the peripheral T cell pool[10,11,20]. Thus, it is important to understand the mechanisms that are required for the maintenance of this peripheral T cell pool. We show that Tcf1[+] cells are indispensable for memory T cell inflation and upholding this subset is therefore critical for CMV-based vaccination approaches.

## Methods

**Ethics statement**. This study was conducted in accordance to the guidelines of the animal experimentation law (SR 455.163; TVV) of the Swiss Federal Government. The protocol was approved by Cantonal Veterinary Office of the canton Zürich, Switzerland (Permit number 146/2014, 114/2017 and 115/2017).

**Mice**. C57BL/6J were purchased from Janvier Elevage and were used as WT mice. The congenic CD45.1 (Ly5.1) mice were bred in house. C57BL/6N-Tg(TcraM38, TcrbM38)329Biat (Maxi) mice express a TCR specific for the MCMV peptide M38$_{316-323}$[8] on the congenic CD45.1 background. $Tcf7^{GFP}$ mice are described[27] and were bred to the P14 TCR transgenic CD45.1[+] mice[62] or the CD45.1[+] TCR transgenic Maxi mouse. CD45.2 $Tcf7^{DTR-GFP}$ P14 mice are described[38]. CD45.1[+] Thy1.1[+] IL12Rβ2[−/−] P14 T cells were generated by crossing the IL12Rβ2[−/−] mice[63] with the congenic Thy1.1 strain and the P14 strain. Thy1.1[+] IFNAR[−/−] P14 mice are described[64]. CCR7[−/−] mice[65] were obtained from the Swiss immunological mouse repository (SwimmR) and were crossed to the TCR transgenic CD45.1[+] Mini mice that express a high avidity TCR β-chain for the M38$_{316-323}$ epitope that pair with endogenous TCR Vα-chains leading to approximatley 10% of the CD8 T cells to be specific for the M38$_{316-323}$ epitope based on M38-tetramer staining[8]. Mice were housed and bred in individually ventilated cages under specific pathogen-free conditions at the Eidgenössische Technische Hochschule (ETH) Hönggerberg. Mice were exposed to a 12:12 h light-dark cycle with unrestricted access to water and food. All mice were between 7–12 weeks at the start of each experiment and were age- and sex-matched.

**Viruses**. MCMV-Δm157 is previously described[66] and referred to as MCMV. For an in vivo infection, 10[6] plaque forming units (PFU) (high dose) or 10[3] PFU (low dose) was administrated intravenously (i.v.). MCMV-ie2-GP33 was obtained from Dr. L. Cicin-Sain, contains the m157 gene and is described[29]. Mice were i.v. infected with 2 × 10[5] PFU MCMV-ie2-GP33. Viral stocks were propagated on M2-10B4 cells. The virus was subsequently purified by ultracentrifugation using a 15% sucrose cushion. Viral titers were determined by standard plaque assay using M2-10B4 cells[67]. LCMV Clone 13 was provided by Dr. R. M. Zinkernagel (University Hospital Zürich, Switzerland) and was propagated on BHK-21 cells at a low multiplicity of infection. 24–48 h post-infection, supernatant was harvested and filtered through a 0.22 μm filter (TPP filtermax) to remove cell debris. To induce a chronic infection, mice were i.v. injected with 2 × 10[6] focus forming units (ffu) LCMV Clone 13.

**Flow cytometry**. Single cell suspensions were prepared from spleen and LNs by meshing the tissue through a 70 μM cell strainer. Inguinal LNs (unless otherwise stated) were used. Erythrocytes were lysed using a hypotonic ammonium-chloride-potassium buffer for 1 min. To collect lungs, mice were perfused with PBS to remove all blood associated cells. For preparation of a single cell suspension, the tissue was cut into small pieces, subsequently incubated with Collegenase I and DNAse I for 45 min, followed by a 30% percoll gradient. Cells were incubated with fluorescently conjugated antibodies for 30 min at 4 °C. Dead cells were excluded using a LIVE/DEAD fixable NEAR-IR staining. For intracellular staining of Tcf1, the FoxP3 kit (invitrogen) was used according to manufacterer's protocol. To determine the proliferation history of Tcf1[+] and Tcf1[−] cells, cells were labelled with cell proliferation dye efluor 450 according to manufacterer's protocol (life technologies). For intracellular cytokines staining, sorted Tcf1[+] or Tcf1[−] P14 T cells were in vitro restimulated for 5 h with 1 μg/ml GP$_{33-41}$ peptide in the presence of 2 μg/ml Brefeldin A. After stimulation, the cell surface was stained for 30 min at 4 °C, after which the cells were washed and fixed overnight with 0.5% PFA. The following day, cells were washed with perm/wash buffer (eBioscience) and cytokines were stained intracellularly for 30 min at 4 °C. The fluorescently conjugated antibodies were diluted in perm/wash buffer. Multi-parametric flow cytometric analysis was performed using LSRII flow cytometer (BD Biosciences) or LSRFortessa (BD Biosciences) with FACSDiva software. Data were analysed using FlowJo software (Tree Star). A representative gating strategy is shown in Supplementary Fig. 7. All antibodies used in this study are indicated in Supplementary Table 1 and are referenced in the reporting summary.

**MHC Class I tetramers**. MHC Class I tetramers for M45$_{985-994}$, GP$_{33-41}$ (both D[b] restited) and M38$_{316-323}$ (K[b] restricted) were produced as described[68] and were conjugated to APC or PE.

**Adoptive transfer and sorting experiments**. P14, Mini or Maxi CD8 T cells were enriched from spleens and LNs by negative selection using the mojosort CD8 T cell isolation kit according to manufacturer's protocol (Biolegend). 10[4] (unless otherwise stated) TCR transgenic cells were adoptively transferred into new hosts that were subsequently infected with MCMV. After at least 4 weeks, Tcf1[+] and Tcf1[−] cells were sorted from spleens and LNs, using a BD FACS Aria sorter. CD4 T cells and B cells were depleted before cell sorting, using biotinilated CD4 and B220 antibodies and Mojosort Streptavidin nanobeads (Biolegend).

**DT, FTY720 treatment and CD62L blocking**. For the experiments where Tcf1[+] P14 T cells were depleted, 4 × 10[3] CD45.2[+] $Tcf7^{DTR-GFP}$ P14 cells were adoptively transferred into CD45.1[+] mice that were subsequently infected with 2 × 10[5] PFU MCMV-ie2-GP33. At day 8, 11, 14 and 17 post-infection, mice received 1 μg DT (Sigma) via an i.p. injection. FTY720 was administrated in the drinking water of mice at a concentration of 5 μg/ml. For CD62L blocking, mice received 100 μg of αCD62L-blocking antibodies i.p. 8 h before and 1, 5, 9, 12, 16 and 19 days post adoptive transfer of Maxi cells.

**Immunofluorescence microscopy**. Spleen and LNs of infected mice were fixed for 1 h in 1% PFA at 4 °C, and subsequently stored in 20% sucroze in PBS overnight at 4 °C. Tissues were embedded in optimum cutting temperature (O.C.T.) medium, frozen in liquid nitrogen and stored at −20 °C until further use. Cryosections of 7 μM were made and air dried. For the staining, slides were re-hydrated in PBS and blocked with 10% rat serum in PBS. Slides were stained with antibodies diluted in PBS containing 1% rat serum, for 1 h in the dark. After subsquent washing with PBS, slides were mounted with Mowiol. Within 1 day, images were acquired on a Visitron confocal system inverse confocal microscope with ×10 magnification. Data were analyzed using the Volocity software.

**TCR sequencing**. Two hundred microlitres of blood was sampled from the saphenous vein of MCMV infected $Tcf7^{GFP}$ mice at indicated timepoints, or by intracardiac bleeding at the terminal timepoint. Blood was collected in heparin-coated microtainers (BD), subsequently centrifuged at 2000g and cell pellet was resuspended in 50 μl PBS. 1.5 ml of Trizol was added and samples were stored at −80 °C until further use. At the terminal timepoint Tcf1[+] and Tcf1[−] M38-specific cells were sorted from spleen, lung, mediastinal LNs and a pool of LNs including inguinal, axillary, lumbar and mesenteric LNs. Cells were lysed in Trizol reagent and stored at −80 °C until further use. RNA was extracted using the Direct-zol RNA MiniPrep kit (Zymo) according to manufacturers instructions.

First strand cDNA was synthesized in a total volume of 20 μl using 11.5 μl of RNA, 0.5 μl oligo(dT) primers (100 mM, life technologies), 1 μl dNTPs (10 mM, life technologies), 1 μl 0.1 M DTT (life technologies), 1 μl RNAsin Plus RNAse inhibitor (10 K, Promega AG), 1 μl Superscript III (200 U/ml, life technologies) and 4 μl 5x Superscript III buffer for 10 min at 50 °C, 10 min at 25 °C and 60 min at 50 °C. Polymerase was inactivated by incubation for 5 min at 94 °C.

TCR sequencing libraries were then prepared in a two-step PCR approach amplifying the TCR-β chain[69] using 19 TRBV forward primers and 1 TRBC reverse primer. All primers are indicated in Supplementary Table 2. The first PCR was performed using Q5 Hotstart Polymerase HiFi (NEB) in a reaction volume of 25 μl with overhang-extended primers under the following conditions (5 × 65 °C, 35 ×

62 °C). PCR product was loaded on a 1% agarose gel and gel pieces were cut out at a product length of around 450 bp. PCR fragments were purified with QIAquick PCR & gel cleanup kit (Qiagen) and eluted in 15 µl.

The entire product was used for the second PCR step (5 × 40 °C, 25 × 65 °C), during which indexed Illumina sequencing adaptors were added. All primers are indicated in Supplementary Table 2. Following another gel purification step, amplicons were eluted in 15 µl of buffer. The quality of the libraries was assessed using a fragment analyzer (Bioanalyzer, Agilent). Libraries passing quality control (uniform product at expected size) were pooled and sequenced on the Illumina MiSeq platform with 2 × 300 base pair.

**TCR analysis.** Paired-end sequencing fastq files were processed using the MiXCR software (v3.0.1) with reads aligned to the built-in murine reference genome[70]. Clonotyping was initially performed on identical nucleotide sequence of the CDR3 region using MiXCR and exported with the preset–full command. Clonotypes containing identical CDR3 amino acid sequences were subsequently merged into the final clone used for the remainder of the analysis. Only those clones aligning to TCR-β constant gene with a productive, in-frame, amino acid CDR3 sequence were included. V-, D-, and J-genes for each clone were determined by selecting the germline segments with the highest alignment scores. Clonal frequencies were calculated by dividing the clone count determined by MiXCR for each clone by the total clone count for each repertoire. When pooling repertoires for a given mouse, the previously calculated clone counts for a given unique clone (based on identical amino acid CDR3) were summed across all organs. Clonal frequencies for pooled repertoires were likewise calculated by dividing the summed clone count for each clone by the total clone count per pooled repertoire.

**Statistical analysis.** Graphpad prism 7.0 software was used to calculate significance between samples. p-values <0.05 were considered as significant. Statistical test is indicated in each figure. Spearman correlation was used to determine correlation between overlapping Tcf1⁺ and Tcf1⁻ clones.

**Reporting summary.** Further information on research design is available in the Nature Research Reporting Summary linked to this article.

## Data availability
The TCR sequencing data files can be found on ArrayExpress with accession E-MTAB-8711 [https://www.ebi.ac.uk/arrayexpress/experiments/E-MTAB-8711/]. The flow cytometry and imaging data supporting the findings are available from the corresponding author upon request. The source data underlying Figs. 1a, b, d, f, h, 2a, d, e, 3c, d, 4c, f, 5c–g, 6b–e, 7a–e, Supplementary Figs. 1b–h, 2b–d, 3a, d, f, g, h, 4a–e, 5a–f, 6a, b are provided in the Source Data File.

## Code availability
Custom R scripts to analyze the TCR sequencing data are available from the corresponding author upon request.

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

## Acknowledgements

This work was supported by the ETH post-doctoral Fellowship program (FEL29 15-2 to SPMW), the Helmut Horten foundation (SPMW), by the SNF (IZHRZ0_180552 to AO and 3100030B_1795709 to WH) and by the DFG (FOR2830 TP5 to LCS). We are grateful to the members of the Oxenius group and the Joller group for helpful discussions.

## Author contributions

S.P.M.W., W.H., A.O. designed the study. S.P.M.W., A.Y., N.B., F.W., N.O., I.S., A.P., J.D.O. performed the experiments. S.P.M.W., A.Y., N.B. analysed the data. S.R., L.C.S., W.H. contributed reagents. S.P.M.W. drafted and wrote the manuscript and all other authors contributed to interpretation of the data and writing of the manuscript. A.O. supervised the project. S.P.M.W., S.R., L.C.S., W.H. and A.O. acquired funding.

## Competing interests

The authors declare no competing interests.
