## [Peer Review File · Nature Communications]

Reviewers' comments:

Reviewer #1 (Remarks to the Author):

In this well-written manuscript, Welten and colleagues show that Tcf1+ CD8+ CMV-tetramer+ central memory T cells in mice are replication competent and feed into a Tcf1- effector T cell pool upon viral reactivation. This is an interesting study, and highly relevant for studies aiming at optimizing experimental therapies based on adoptive transfer of CMV-specific CD8+ T cells. Even if the first sections of results may appear a bit confirmatory to previous studies as for example referred to in refs 24 – 27 and 32, the authors went the extra mile 1) to confirm their hypothesis by Tcrb sequencing of Tcf1+ and Tcf1- T cell pools and showing a strong overlap in the tet+ Tcf1+ and tet+ Tcf- fractions, and 2) to test the relevance of Tcf1+ CD8+ T cells for maintaining memory inflation by employing a genetic model for the specific depletion of Tcf1+ T cells. Together, the manuscript should be suitable and timely for the readership of nature communications.

A few rather minor concerns should be addressed before publication:

- in the introduction, sentences in line 53 and 42 are somewhat conflicting
- line 256 the call out of "1 day after .. " is fully correct, but readers may have to read this part twice looking at the figure labeled day 9
- more critical is the seemingly arbitrary gating on loss of CPD in figures 4b versus 4D and S3C versus S3E, please clarify.

Reviewer #2 (Remarks to the Author):

This is very elegant and well executed work. It established that persisting and inflating T cell responses in chronic CMV infections are maintained by similar mechanism as chronic LCMV infection. Particularly convincing are the transfer experiments which show that proliferative capacity is tightly linked to Tcf1 expression and the elegant DTX mediated depletion experiments. Here the administration of DTX causes in a depletion of the Tcf1 expressing population which resulted in a failure to maintain inflating T cell response to CMV. Beyond this conceptual very important demonstration, the authors establish interesting particularities of these cells which are of key importance to the field.

These include that:

- Tcf1+ cells are formed with similar kinetics in inflating and non-inflating response
- Tcf1+ progenitor cells are found in much higher numbers in the lymph nodes and at much lower frequencies in the spleen.
- Tcf1+ expression and thus cellular maintenance is linked to cells with a central memory phenotype
- Pro-inflammatory cytokines negatively regulate Tcf1 expression

In General the data in the paper is easily interpretable and well characterized. Moreover, the authors did a great job of using multiple setups to sustain their core conclusion, making the data very convincing. I only have a few minor points that should be addressed upon revision:

Minor points:

- Figure 1I

Why is there such a discrepancy between the %Tcf1+ and %gfp reporter positive cells in the lungs (17% versus 5%)?

- Line 145-148 and Supplementary Figure 2:

The Tcf1+ look very similar to the cells identified by in LCMV clone-13 infections. Yet, I assume they will phenotypically be very different. It would be very informative if the phenotypic characterization of the population could be expanded, i.e. do they express PD-1 or other inhibitory marker, what is their cytokine expression profile?

- Line 170-172 and Figure 3D

The data plots show Tcf1 MFI data. This seems inappropriate as the entire population does not express Tcf1. Instead, illustrating the frequency of Tcf1+ positive cells among total cells would be better suited.

- Supplementary Figure 4C-D

What is the difference of WT vs CCR7 ko data in the blood? Throughout the manuscript the authors show data for the blood. It would be nice to show this, especially since these cells cannot home to lymphoid regions.

Other comments:

- Figure 5D, I would suggest to write "DT" instead of "+DT" (or include "+vehicle control") above the bar that indicates the application time of DT. The current label confuses as only one group received DT.

- Figure 5C, The frequencies of gfp+ cells in the spleen, blood, and lung plots are hard/impossible to read in the present form that shows gfp+ and gfp- cells in one bar. Maybe take them out or show the bars for gfp- and gfp+ cells separately.

- Figure 7, It would be helpful to indicate that Figure 7b show the frequency of clones in the Tcf1- population which can be found in the Tcf1+ population and not vice versa.

- Instead of using "Nr" to denote number in your figures I would consider using # or just using the word number.

- Figure 1D ,F: Please add statistics.

- Line 998: This sentence is slightly confusing. I would consider rewording it.

- Line 1047 and 1066: Should be - Bar graph shows or Bar graphs show

- Line 1063: should be mean +/- SEM

Reviewer #3 (Remarks to the Author):

The manuscript Welten et al investigates a T cell subset that they believe is responsible for maintain inflationary T cells in a murine model of murine Cytomegalovirus infection (MCMV). Then manuscript builds on observations from this laboratory that a small Tcm population is enriched in the LN and has a higher proliferative capacity responding to MCMV reactivation. In this new work they investigate the role of a previously described transcription factor (Tcf1) important for Tcm development and already shown to have a role in the LCMV model system expressed by a subset of T cells which self-renewal capacity. As such the group investigate the role of Tcf1 in the MCMV model system of inflationary T cells.

The authors have used CMV in the manuscript title and introduction when in fact all of the data concerns MCMV and the introduction with regard to inflation is based almost entirely on MCMV observations. The situation with memory inflation in HCMV is not anywhere near as clearly established and this should be reflected in the title, introduction and the discussion.

Figure 1

Lines 104-106 I find the statements confusing - the results show that while Tcf1+ cells either M38 or M45 are low in spleen/lung/blood they are significantly increased in LN and in both M38 or M45 specific T cell populations and cannot really be described as hardly detected?

Lines 112-113 I also find confusing while M38 specific cells Tcf1+ are at a higher percentage in LN compared to other sites, M45 Tcf1+ cells are also present, are you suggesting that although both specificities are in LN and express Tcf1 that they are phenotypically different at this site and only the M38 are Tcm - could you please clarify what you mean and the significance of the result.

Lines 114-118 It will not be obvious to some readers that MCMV-IE2 specific cells are inflationary nor that insertion of another antigenic peptide, in this case LCMV gp33 into this region, would then cause these specific T cells to GP-33 to also be inflationary, this should be explained as the P14 transgenics are subsequently used in other parts of the manuscript.

Lines 131-133 I again find the conclusion confusing, Tcf1 is expressed by both M38 and M45 inflationary and non-inflationary MCMV specific T cells in the LN, could you please clarify the point you are trying to make. As mentioned earlier are both specificities in the LN Tcm?

Figure 2

Using the transgenic P14 model the data in this figure phenotypes the cells and shows that they are Tcm and they are in a location consistent with Tcm cells. Results are discussed for CxCR5 expression and in addition inoculum dose. It is difficult to understand the rationale for using this model system. Why not show the phenotype of the M38 and M45 specific cells? Why is a discussion of inoculum dose and the degree of memory inflation relevant to the story that the authors are developing?

Figure 3

The data in this figure recapitulates previously published work showing that IL-12 and type 1 IFNs regulate Tcf1. The system used is again P14 transgenic, given the paper is trying to understand inflationary v non-inflationary responses why use this system could the experiments not have been done looking at the M45 and M38 specific T cells.

Figure 4

Utilizes cell sorting of Tcf1 GFP positive and negative cells labelled with a CPD to track proliferation to determine proliferation potential, the results show that Tcf1+ cells proliferate on antigen encounter and form the Tcf1- population.

Again, I question why the experiment could not be done with M38 and M45 specific T cells. However, another transgenic mouse system (Maxi) is used at this point for M38 specificity. This supports the results of the P14 system and is shown as supplementary data. Why not show the M38 results as that reflects the data from the first figure and put the P14 results in supplementary?

I am still left asking myself the question what the difference is between the M38 and M45 Tcf1+ cells from LN, phenotypically and now if the M45 cells can also proliferate to antigen but then give rise to Tcf1+ cells in the periphery. One would assume that the peripheral non-inflationary cells

also need to be maintained they do not expand after contraction but they don't contract completely do these cells also not need to be homeostatically replaced? This seems to me an important question and should be experimentally addressed.

Figure 5

Why is the "Inflation" of Tcf1- P14 cells in 5d so small in comparison to other experiments? It would appear that the DT treated and untreated mice T cells contract to almost the same degree, there is then a doubling .2 to .4% of these cells in the untreated mouse which are then held at steady state?

Figure 6 and Figure 7

Utilizing TcR sequencing shows that there is a greater clonal diversity in the M38 Tcf1+ cells as compared to Tcf1- cells but that the negative cells clonally overlap with the Tcf-1 negative peripheral population. The authors do not speculate on the M38 Tcf1 specific T cell clonotypes that do not make it into the peripheral Tcf1- pool. What do these cells do if upon antigen encounter? Do they proliferate but maintain Tcf-1 and LN residency?

Point-by-point reply**"Tcf1⁺ cells are required to maintain the inflationary T cell pool upon MCMV infection"**

We would like to thank all three reviewers for their critical, constructive and very valuable comments. We believe that we can address all the raised concerns as detailed below.

Specific replies to the individual reviewers:

Reviewer #1 (Remarks to the Author):

In this well-written manuscript, Welten and colleagues show that Tcf1⁺ CD8⁺ CMV-tetramer⁺ central memory T cells in mice are replication competent and feed into a Tcf1⁻ effector T cell pool upon viral reactivation. This is an interesting study, and highly relevant for studies aiming at optimizing experimental therapies based on adoptive transfer of CMV-specific CD8⁺ T cells.

Even if the first sections of results may appear a bit confirmatory to previous studies as for example referred to in refs 24 – 27 and 32, the authors went the extra mile 1) to confirm their hypothesis by Tcrb sequencing of Tcf1⁺ and Tcf1⁻ T cell pools and showing a strong overlap in the tet⁺ Tcf1⁺ and tet⁺ Tcf1⁻ fractions, and 2) to test the relevance of Tcf1⁺ CD8⁺ T cells for maintaining memory inflation by employing a genetic model for the specific depletion of Tcf1⁺ T cells. Together, the manuscript should be suitable and timely for the readership of nature communications. A few rather minor concerns should be addressed before publication:

- *in the introduction, sentences in line 53 and 42 are somewhat conflicting*

We thank the reviewer for addressing this point. We have changed replication deficient in line 52 into spread deficient. This change is highlighted in the revised manuscript.

"Although one has to be careful with using live viral vectors for vaccination purposes, attenuated CMV strains that are **spread**-deficient also induce memory T cell inflation, providing a safer alternative."

- *line 256 the call out of "1 day after .. " is fully correct, but readers may have to read this part twice looking at the figure labeled day 9*

We thank the reviewer for addressing this point. We have made it clearer in the text that 1 day after DT treatment represents day 9 post-infection. Line 240 of the manuscript.

- *More critical is the seemingly arbitrary gating on loss of CPD in figures 4b versus 4D and S3C versus S3E, please clarify.*

We thank the reviewer for raising this concern. The gating in Figure 4B and Supplemental Figure 3C is set so that all cells that have divided at least once are included. This explanation is included in the manuscript and can be found in line 175. The cells that have divided once likely reflect homeostatic proliferation as these cells are also found in naïve recipients. This gate also includes all cells that have undergone multiple rounds of division, which is likely driving by antigen encounter. The gating in Figure 4D and Supplemental Figure 3E is set to select only the cells that have undergone multiple rounds of division, indicated by the out-dilution of the CPD, and have upregulated KLRG1. These cells are only found in MCMV infected hosts and not in naïve hosts and represent a population that arises due to antigen encounter.

Reviewer #2 (Remarks to the Author):

This is very elegant and well executed work. It established that persisting and inflating T cell responses in chronic CMV infections are maintained by similar mechanism as chronic LCMV infection. Particularly convincing are the transfer experiments which show that proliferative capacity is tightly linked to Tcf1 expression and the elegant DTX mediated depletion experiments. Here the administration of DTX causes in a depletion of the Tcf1 expressing population which resulted in a failure to maintain inflating T cell response to CMV. Beyond this conceptual very important demonstration, the authors establish interesting particularities of these cells which are of key importance to the field.

These include that:

- *Tcf1+ cells are formed with similar kinetics in inflating and non-inflating response*
- *Tcf1+ progenitor cells are found in much higher numbers in the lymph nodes and at much lower frequencies in the spleen.*
- *Tcf1+ expression and thus cellular maintenance is linked to cells with a central memory phenotype*
- *Pro-inflammatory cytokines negatively regulate Tcf1 expression*

In general, the data in the paper is easily interpretable and well characterized. Moreover, the authors did a great job of using multiple setups to sustain their core conclusion, making the data very convincing. I only have a few minor points that should be addressed upon revision:

Minor points:

- Figure 11

- *Why is there such a discrepancy between the %Tcf1⁺ and %gfp reporter positive cells in the lungs (17% versus 5%)?*

We thank the reviewer for addressing this question. We have also noticed this difference. The Tcf7^{GFP} reporter mice contain a BAC construct that reports GFP expression under the control of the *Tcf7* promoter. One explanation for the discrepancy between the Tcf1 protein staining and the GFP signal in the lungs might be a difference in half-life of the Tcf1 and the GFP proteins. The turnover of Tcf1 protein is very fast. The half-life is estimated to be around a few hours [1], whereas the half-life of GFP is longer and estimated to be around 26 hours [2]. The cells in the lungs that are still GFP⁺ but do not express the Tcf1 protein would therefore represent cells that have recently downregulated Tcf1. Another explanation might be a different epigenetic regulation of transcription of the large BAC construct compared to the chromosomal *Tcf7* gene.

- Line 145-148 and Supplementary Figure 2:

- *The Tcf1⁺ look very similar to the cells identified by in LCMV clone-13 infections. Yet, I assume they will phenotypically be very different. It would be very informative if the phenotypic characterization of the population could be expanded, i.e. do they express PD-1 or other inhibitory marker, what is their cytokine expression profile?*

We thank the reviewer for addressing this question. We have performed an additional experiment where we determined the expression of different co-inhibitory receptors and other cell surface markers on splenic Tcf1⁺ and Tcf1⁻ P14 T cells in MCMV infection and compared this to a setting in which mice were chronically infected with LCMV. This new data is shown in Figure A and is included in Supplementary Figure 2C in the revised version of the manuscript.

LCMV-specific CD8 T cells are characterized by expression of the co-inhibitory receptors PD-1, Tim3, Lag3, and CD39. The majority of LCMV-specific T cells expresses PD-1, irrespective of Tcf1 expression. However, in MCMV infection no PD-1 expression is detected. Furthermore, Lag3 and Tim3 are only expressed by Tcf1⁻ cells in the LCMV setting, but these markers are hardly expressed during MCMV infection. CD39, however, is found on MCMV-specific CD8 T cells and is mostly expressed by Tcf1⁻ cells.

Ly108 (Slamf6) expression is in LCMV infection correlated with Tcf1⁺ expression. Also in MCMV infection more Tcf1⁺ cells express Ly108, however, not all Tcf1⁺ co-express this marker.

Inflatory T cells are characterized by a low cell surface expression of the costimulatory molecule CD27 [3]. There is a small population that retains a higher expression of CD27, and these cells express Tcf1. In LCMV infection, both Tcf1⁺ and Tcf1⁻ cells express CD27.

In addition, we checked in LCMV infection expression of CD62L, CD127, CX3CR1 and KLRG1, as these markers were shown in Figure 2 of the manuscript in MCMV infection. In both settings, CD62L and CD127 are expressed pre-dominantly by Tcf1⁺ cells and CX3CR1 and KLRG1 by Tcf1⁻ cells. However, in general the percentages of virus-specific cells that expresses these markers is higher in MCMV infection.

Thus, the most striking difference in the Tcf1⁺ population is that these cells express PD-1 in LCMV infection, but not in MCMV infection.

We also determined the IFN γ and IL-2 production of Tcf1⁺ and Tcf1⁻ cells by restimulating these cells *in vitro* with GP₃₃₋₄₁ peptide. As we cannot exclude that Tcf1 expression is affected upon *in vitro* exposure to cognate antigen, we sorted GFP⁺ and GFP⁻ P14 T cells from the spleen of MCMV- and LCMV-infected mice. This new data is shown in Figure A and included in Supplementary Figure 2D in the revised version of the manuscript. Tcf1⁺ cells are better IL-2 producers in both MCMV and LCMV infection, however, a significantly larger proportion produces IL-2 in MCMV compared to LCMV infection. Whereas Tcf1⁻ cells are better IFN γ producers in LCMV infection, this difference is not evident in MCMV infection. In general, a higher proportion of P14 T cells produce cytokines in MCMV infection as compared to LCMV infection.

Figure A: Phenotypical characterization of Tcf1⁺ cells in MCMV and LCMV infection.

10^4 CD45.1⁺ Tcf7^{GFP} P14 cells were transferred into WT mice that were subsequently infected with 2×10^5 PFU MCMV-*ie2*-GP33 or with 2×10^6 ffu LCMV-Clone 13. (A) The expression of several cell surface markers is shown on splenic Tcf1⁺ and Tcf1⁻ P14 cells, 21 days post-infection. Mean is indicated. (B) Tcf1⁺ and Tcf1⁻ P14 cells were sorted from the spleen of MCMV- and LCMV-infected mice. Shown are flow cytometry plots for the intracellular IFN γ and IL-2 production after restimulation with GP₃₃₋₄₁ peptide and the percentage of P14 cells that produces IFN γ or IL-2 as mean \pm SEM. Data are from 1 experiment, each dot is an individual mouse, n=5. *p<0.05; statistical significance was determined using the non-parametric Mann-Whitney test.

- Line 170-172 and Figure 3D

- The data plots show Tcf1 MFI data. This seems inappropriate as the entire population does not express Tcf1. Instead, illustrating the frequency of Tcf1⁺ positive cells among total cells would be better suited.

We thank the reviewer for addressing this point. We have replaced Figure 3D with a new bar graph that shows the percentage of P14 cells that expresses Tcf1. Similar as the previous bar graph that displayed the MFI of Tcf1, it is evident that a larger percentage of IL12R β 2^{-/-} and IFNAR^{-/-} P14 T cells expresses Tcf1 as compared to WT cells. This effect is most pronounced in the lymph node and in the spleen. Of note, it is striking that the expression of Tcf1 in P14 T cells on day 8 post infection is higher than the expression observed for M45- and M38-specific CD8 T cell at this time point. We believe this is due to antigen abundance, as the GP33 epitope is inserted in the *ie2* gene, which is less abundantly expressed in the acute phase of infection than M45 and M38.

- Supplementary Figure 4C-D

- What is the difference of WT vs CCR7 ko data in the blood? Throughout the manuscript the authors show data for the blood. It would be nice to show this, especially since these cells cannot home to lymphoid regions.

In Supplementary Figure 4C and 4E of the revised manuscript we have included additional data regarding the CCR7 KO Mini cells in the blood. Supplementary Figure 4C shows the percentage of Mini cells amongst total lymphocytes at day 77 post infection. It is evident that comparable to the spleen, LN and lungs, less Mini cells are accumulating in the blood. Supplementary Figure 4E shows that the Mini cells that are found in the blood have a diminished effector memory phenotype. Together these data show that homing to secondary lymphoid organs is critical for T cell accumulation to occur in the periphery.

Other comments:

- *Figure 5D, I would suggest to write “DT” instead of “+DT” (or include “+vehicle control”) above the bar that indicates the application time of DT. The current label confuses as only one group received DT.*

We thank the reviewer for addressing this point. We have changed +DT into DT in Figure 5D.

- *Figure 5C, The frequencies of gfp+ cells in the spleen, blood, and lung plots are hard/impossible to read in the present form that shows gfp+ and gfp- cells in one bar. Maybe take them out or show the bars for gfp- and gfp+ cells separately.*

We thank the reviewer for addressing this point. We have divided Figure 5C into two new figures. Figure 5C shows the GFP⁺ and GFP⁻ population in the lymph nodes, as at this site there is still a small Tcf1⁺ population detected in mice that did not receive DT treatment. The new Figure 5D shows only the Tcf1⁻ population in the spleen, lungs and the blood as at these sites hardly any Tcf1⁺ cells are found at this time.

- *Figure 7, It would be helpful to indicate that Figure 7b show the frequency of clones in the Tcf1- population which can be found in the Tcf1+ population and not vice versa.*

We thank the reviewer for addressing this point. We have indicated this in the figure legend of Figure 7B. This can be found in line 923-924 of the manuscript.

- *Instead of using “Nr” to denote number in your figures I would consider using # or just using the word number.*

We thank the reviewer for addressing this point. In all the figures Nr is replaced by #.

- *Figure 1D, F: Please add statistics.*

We thank the reviewer for addressing this point. The statistics has been added to Figure 1D and 1F.

- *Line 998: This sentence is slightly confusing. I would consider rewording it.*

We thank the reviewer for addressing this point. We have changed the sentence.

- Line 1047 and 1066: Should be - Bar graph shows or Bar graphs show

We thank the reviewer for pointing out this mistake. It has been changed in the revised version of the manuscript. These changes can be found in line 1012 and line 1033.

- Line 1063: should be mean +/- SEM

We thank the reviewer for addressing this point. We checked all the figures with their corresponding legends and made sure everything is correctly indicated.

Reviewer #3 (Remarks to the Author):

The manuscript Welten et al investigates a T cell subset that they believe is responsible for maintain inflationary T cells in a murine model of murine Cytomegalovirus infection (MCMV). Then manuscript builds on observations from this laboratory that a small Tcm population is enriched in the LN and has a higher proliferative capacity responding to MCMV reactivation. In this new work they investigate the role of a previously described transcription factor (Tcf1) important for Tcm development and already shown to have a role in the LCMV model system expressed by a subset of T cells which self-renewal capacity. As such the group investigate the role of Tcf1 in the MCMV model system of inflationary T cells.

The authors have used CMV in the manuscript title and introduction when in fact all of the data concerns MCMV and the introduction with regard to inflation is based almost entirely on MCMV observations. The situation with memory inflation in HCMV is not anywhere near as clearly established and this should be reflected in the title, introduction and the discussion.

We thank the reviewer for addressing his concern regarding the adequate description for the use of MCMV. We have changed the title and explained in the introduction that we are using MCMV as our model. Moreover, the discussion contains a paragraph speculating on the role of Tcf1⁺ cells during HCMV infection (line 370-383).

"We observed clonal overlap between Tcf1⁺ and Tcf1⁻ cells, with a larger TCR repertoire diversity in the Tcf1⁺ population. Clonal TCR analysis of HCMV-specific CD8 T cells did not show much overlap between central memory (IL7Rα⁺) and effector memory T cells (IL7Rα⁻) in the lymph nodes and blood, as many clones were only found in one compartment [4]. However, the spleen was not included in this study and therefore clones might have been underrepresented. There is quite some heterogeneity in the frequency and tissue distribution

of HCMV-specific CD8 T cells amongst individuals [5] making it necessary to include multiple tissues to get the full spectrum of clonotypes. If a subset of HCMV-specific inflationary CD8 T cells expresses Tcf1 remains to be investigated, but is likely based on the following observations. HCMV-specific CD8 T cells isolated from the lymph node have a higher proliferative capacity than similar cells obtained from the blood [6]. Increased Tcf1 expression is found in memory CD8 T cells from human lymph nodes compared to other tissues and these memory cells a higher clonal diversity [6]. Furthermore, Tcf1 marks human CD8 T cells capable of self-renewing and upon proliferation these Tcf1⁺ cells give rise to Tcf1⁻ cells [7, 8]."

Figure 1

- *Lines 104-106 I find the statements confusing - the results show that while Tcf1⁺ cells either M38 or M45 are low in spleen/lung/blood they are significantly increased in LN and in both M38 or M45 specific T cell populations and cannot really be described as hardly detected?*

We thank the reviewer for addressing this point. We have changed the text in the manuscript into:

"Also in the spleen, lungs and lymph node, **only a small fraction of both M45- and M38-specific T cells expressed Tcf1** 8 days post-infection (Fig. 1C, D), although a slightly higher percentage of Tcf1-expressing cells was found in the lymph node."

- *Lines 112-113 I also find confusing while M38 specific cells Tcf1⁺ are at a higher percentage in LN compared to other sites, M45 Tcf1⁺ cells are also present, are you suggesting that although both specificities are in LN and express Tcf1 that they are phenotypically different at this site and only the M38 are Tcm – could you please clarify what you mean and the significance of the result.*

We thank the reviewer for addressing this point. We do not suggest that M45- and M38-specific CD8 T cells are phenotypically different in the lymph nodes. On the contrary, the lymph node is the only site where these two populations have more phenotypical overlap after resolution of acute MCMV infection. M38-specific T cells have a predominant effector memory phenotype in the periphery, whereas in the lymph node M38-specific T cells have a more central memory phenotype. M45-specific T cells have independent of their localization (periphery or lymph nodes) predominantly a central memory phenotype. This dominant effector memory phenotype of M38-specific CD8 T cells in non-lymphoid tissues is due to encounter of viral antigen in the persistent/latent phase of MCMV infection. M38 antigen is presented by latently infected non-hematopoietic cells and this drives the differentiation of the M38-specific central memory cells into a

more effector-like phenotype. M45 antigen is not presented in the persistent/latent phase of MCMV infection. Therefore the M45-specific CD8 T cells are not reactivated and remain with a central memory phenotype.

As we show later in the manuscript that T_{CM} cells express Tcf1 and these Tcf1⁺ cells feed into the peripheral pool of inflationary T cells, it is of significance to mention where these cells most prominently residing.

- *Lines 114-118 It will not be obvious to some readers that MCMV-*ie2* specific cells are inflationary nor that insertion of another antigenic peptide, in this case LCMV gp33 into this region, would then cause these specific T cells to GP-33 to also be inflationary, this should be explained as the P14 transgenics are subsequently used in other parts of the manuscript.*

We thank the reviewer for addressing this point. The following sentence is written in the manuscript which describes that insertion of the GP33 epitope into the *ie2* locus induces a GP33-specific CD8 T cells response with inflationary characteristics. Line 102-105.

"CD45.1⁺ P14 Tcf7^{GFP} cells were transferred into CD45.2⁺ WT mice that were subsequently infected with an MCMV variant expressing the GP₃₃₋₄₁ epitope in the *ie2* locus (MCMV-*ie2*-GP33), upon which a GP33-specific T cell response with inflationary characteristics is elicited that accumulates in the blood [9] (Supplementary Fig. 1E)."

We show in Supplementary Figure 1E the kinetics of the P14 cells upon an MCMV-*ie2*-GP33 infection. This figure shows that P14 T cells accumulate in the blood after resolution of acute MCMV infection. In Supplementary Fig 1G, the endogenous GP33-specific response is plotted in the blood in time. Also in this plot it is evident that GP33-specific cells accumulate in the blood. In the manuscript, we also refer to a paper in which the MCMV-*ie2*-GP33 virus was first described [9]. In this publication a graph is included that shows the kinetics and the phenotype of the GP33-specific response, resembling an inflationary T cell response indicated by accumulation of KLRG1-expressing GP33-specific CD8 T cells in time. This data is comparable to what we show in Supplementary Figure 1E and 1G.

- *Lines 131-133 I again find the conclusion confusing, Tcf1 is expressed by both M38 and M45 inflationary and non-inflationary MCMV specific T cells in the LN, could you please clarify the point you are trying to make. As mentioned earlier are both specificities in the LN Tcm?*

We thank the reviewer for addressing this point. As explained above, the majority of the inflationary M38-specific cells in the lymph node has a T_{CM} phenotype and expresses Tcf1. In the periphery these cells have a more effector phenotype and do not express Tcf1. Thus the only site where M38-specific cells express Tcf1 is the lymph node. The majority of M45-specific cells have independent of their localization a central memory phenotype and express Tcf1. Thus, in the lymph node both M45-specific and M38-specific cells have a central memory phenotype and express Tcf1, whereas these cells are phenotypically more different in the periphery.

Figure 2

- *Using the transgenic P14 model the data in this figure phenotypes the cells and shows that they are Tcm and they are in a location consistent with Tcm cells. Results are discussed for CxCR5 expression and in addition inoculum dose. It is difficult to understand the rationale for using this model system. Why not show the phenotype of the M38 and M45 specific cells? Why is a discussion of inoculum dose and the degree of memory inflation relevant to the story that the authors are developing?*

We thank the reviewer for addressing these questions. As explained above, using P14 T cells in combination with an MCMV-*ie2*-GP33 infection induces a P14 T cell response with inflationary characteristics. We have shown in the manuscript data for the phenotype of the P14 T cells, but the phenotypical data for the TCR transgenic Maxi cells and the endogenous M38-specific CD8 T cells show very similar findings. This is indicated in Figure B of the point by point reply. A larger fraction of the Tcf1⁺ population expresses CD127 and CD62L as compared to the Tcf1⁻ cells, and a larger proportion of the Tcf1⁻ cells expresses KLRG1. These differences are observed irrespective if P14 T cells upon MCMV-*ie2*-GP33 infection are analyzed, or whether TCR transgenic Maxi or endogenous M38-specific CD8 T cells in combination with MCMV- Δ m157 are used.

Due to the question of reviewer 2 regarding differences in phenotype between Tcf1⁺ cells in MCMV and LCMV infection, we have included in Supplementary Figure 2 in the revised version of the manuscript new plots that show the expression of different markers on P14 T cells in LCMV infection. Therefore we have decided to leave the phenotypical data for the P14 T cells upon MCMV-*ie2*-GP33 in the manuscript as by using cells with a similar TCR the phenotypical data of P14 T cells in MCMV infection can directly be compared to the LCMV setting.

Memory inflation is influenced by factors from both the virus and the host perspective. From the host perspective, this is related to the number of T_{CM} cells that are able to sense viral reactivation events

[10]. From the virus side, this is influenced by the number of viral reactivation events which is linked to the latent number of viral genomes and the initial inoculum dose [11]. It has been shown before that the viral inoculum dose impacts on the degree of memory inflation [12], with a higher dose leading to a higher accumulation of effector-like cells in the periphery. As we observed that Tcf1 expression is correlated to a central memory phenotype, and the dose impacts the proportion of T_{CM} cells, we expected to find a larger fraction of Tcf1⁺ when a lower inoculum dose of MCMV was used. This is what we found and is shown in Figure 2E of the manuscript.

Figure B: Tcf1⁺ cells express markers associated with a central memory phenotype.

(A) CD45.1⁺ Tcf7^{GFP} P14 cells were adoptively transferred into WT mice that were subsequently infected with 2×10^5 PFU MCMV-ie2-GP33. The cell surface expression of CD62L, CD127 and KLRG1 was determined on Tcf1⁺ and Tcf1⁻ P14 T cells at day 35 post infection. (These plots are also found in Figure 2A of the manuscript) (B) CD45.1⁺ Tcf7^{GFP} Maxi cells were adoptively transferred into CD45.2⁺ hosts that were subsequently i.v. infected with 10^6 PFU MCMV Δ m157. The cell surface expression of CD62L, CD127 and KLRG1 was determined on Tcf1⁺ and Tcf1⁻ Maxi cells at day 35 post infection. (C) Mice were i.v. infected with 10^6 PFU MCMV Δ m157. The cell surface expression of CD62L, CD127 and KLRG1 was determined on Tcf1⁺ and Tcf1⁻ endogenous M38-specific cells at day 58 post infection. All bar graphs

represent mean \pm SEM and each dot represents an individual mouse. * $p < 0.05$; statistical significance was determined using the non-parametric Mann-Whitney test.

Figure 3

- *The data in this figure recapitulates previously published work showing that IL-12 and type I IFNs regulate Tcf1. The system used is again P14 transgenic, given the paper is trying to understand inflationary v non-inflationary responses why use this system could the experiments not have been done looking at the M45 and M38 specific T cells.*

We thank the reviewer for raising his concerns, however, we are not trying to understand the role of Tcf1 in inflationary versus non-inflationary responses, but we want to understand which subset of cells feeds into the inflationary T cell population. We specifically wanted to assess the role of type I IFNs and IL-12 signaling on Tcf1 expression on CD8 T cells. We cannot use a host that is completely deficient for either receptor as other cells might also be affected when they cannot respond to these pro-inflammatory cytokines. Therefore we cannot assess endogenous M45- and M38-specific T cell response and we required IL-12R β 2 or IFNAR deficient TCR transgenic T cells that could be used for adoptive transfer. There is no Maxi mouse available on the IL-12R β 2 or IFNAR deficient background. In addition, there is no M45-specific TCR transgenic mouse available according to our knowledge.

Figure 4

- *Utilizes cell sorting of Tcf1 GFP positive and negative cells labelled with a CPD to track proliferation to determine proliferation potential, the results show that Tcf1+ cells proliferate on antigen encounter and form the Tcf1- population. Again, I question why the experiment could not be done with M38 and M45 specific T cells. However, another transgenic mouse system (Maxi) is used at this point for M38 specificity. This supports the results of the P14 system and is shown as supplementary data. Why not show the M38 results as that reflects the data from the first figure and put the P14 results in supplementary?*

We thank the reviewer for addressing this point. It is more convenient to perform these experiments using TCR transgenic T cells as the number of cells we require is quite high and many donor mice would be needed if we would sort for endogenous M38-specific T cells.

Performing a similar experiment with M45-specific T cells would not answer the question we are trying to address with this experiment; Do Tcf1⁺ cells feed into the peripheral inflationary T cell pool? In order for memory inflation to occur, antigen presentation is critical. M45-specific cells do not elicit an inflationary response as the M45-epitope is not presented during persistent/latent MCMV infection and therefore using M45-specific cells for adoptive transfer would not give rise to a peripheral pool of KLRG1⁺ inflationary T cells. This is also shown by Snyder et al.[13] (Figure 4 of reference 12) where adoptive transfer of either CFSE labelled M45- or M38-specific cells into infection-matched recipients leads to a larger population that has outdiluted the CFSE for M38-specific cells than for M45-specific cells. Furthermore, the M45-specific cells divide with a comparable rate in naïve hosts, reflecting homeostatic proliferation and not antigen driven proliferation.

We have decided to leave the P14 data as the main figure as also the experiment in which Tcf1⁺ cells are depleted using diphtheria toxin is performed with P14 T cells. We agree with the reviewer that the results of the Maxi cells resemble more the natural situation as Maxi cells recognize a natural MCMV epitope. However, similar results are obtained using either P14 or Maxi cells. As we show in Figure A of the point by point reply, P14 T cells, Maxi cells and endogenous M38-specific cells have similar phenotypical characteristics in MCMV infection so we believe that studying the P14 response upon MCMV-*ie2*-GP33 is a good model for understanding memory inflation.

- *I am still left asking myself the question what the difference is between the M38 and M45 Tcf1⁺ cells from LN, phenotypically and now if the M45 cells can also proliferate to antigen but then give rise to Tcf1⁺ cells in the periphery. One would assume that the peripheral non-inflationary cells also need to be maintained they do not expand after contraction but they don't contract completely do these cells also not need to be homeostatically replaced? This seems to me an important question and should be experimentally addressed.*

We thank the reviewer for addressing this point. To answer this question, it is important to make the distinction between antigen-driven and homeostatic proliferation. In our manuscript, we are investigating antigen-driven proliferation that maintains the inflationary T cell pool at high frequency. As discussed above, for memory inflation to occur, antigen presentation is required. The M45 antigen is not presented during viral latency. If M45 antigen would be presented we would expect antigen-driven proliferation and in that case also the Tcf1⁺ cells of the M45 population would respond and give rise to the Tcf1⁺ M45 cells.

M45-specific T cells follow the kinetics of a conventional T cell response. It has been shown before that Tcf1 is required for differentiation and longevity of memory T cells [14, 15]. In experimental mouse models, Tcf7^{-/-} CD8 memory T cells undergo progressive loss in time due to a diminished responsiveness to IL-15 driven homeostatic proliferation [15]. The pool of M45-specific cells is, like all conventional memory T cells, maintained by homeostatic proliferation [13]. Thus, for the maintenance of this population Tcf1 expression is required and also the Tcf1⁺ cells will divide but will give rise to Tcf1⁺ cells.

Figure 5

- Why is the “Inflation” of Tcf1- P14 cells in 5d so small in comparison to other experiments? It would appear that the DT treated and untreated mice T cells contract to almost the same degree, there is then a doubling .2 to .4% of these cells in the untreated mouse which are then held at steady state?

We thank the reviewer for addressing this question. The difference in the degree of memory inflation is due to the different amount of TCR transgenic cells that were transferred before infection. Figure C shows a similar experiment as shown in Figure 5D of the manuscript only more P14 Tcf7^{DTR-GFP} T cells were transferred before infection, 10^4 compared to 4×10^3 in Figure 5D. Upon transfer of more TCR transgenic cells, the percentage of cells found in the blood throughout MCMV infection is higher as well. We have shown before that the precursor frequency of MCMV-specific CD8 T cells impact on the degree of memory inflation [10]. The reason that different numbers of P14 T cells were used between experiments was due to the different cell numbers that were obtained after CD8 T cell enrichment.

Figure C: Depletion of Tcf1⁺ cells hampers memory inflation.

Experimental setup: 10⁴ CD45.2⁺ Tcf7^{DTR-GFP} P14 T cells were adoptively transferred in CD45.1 hosts that were subsequently infected with 2 × 10⁵ PFU MCMV-ie2-GP33. On day 8, 11, 14 and 17 post infection one group of mice received 1 μg DT via an i.p. injection. The percentage of Tcf1⁺ P14 T cells (GFP⁺) in the CD8 T cell pool is shown in the blood in mice with or without DT treatment.

Figure 6 and Figure 7

- *Utilizing TcR sequencing shows that there is a greater clonal diversity in the M38 Tcf1⁺ cells as compared to Tcf1⁻ cells but that the **negative** cells clonally overlap with the Tcf-1 negative peripheral population. The authors do not speculate on the M38 Tcf1 specific T cell clonotypes that do not make it into the peripheral Tcf1⁻ pool. What do these cells do if upon antigen encounter? Do they proliferate but maintain Tcf-1 and LN residency?*

We thank the reviewer for addressing this question, but we do not fully understand the question of the reviewer as we do not show that the Tcf1⁻ cells overlap with the peripheral Tcf1⁻ population. We suspect the reviewer is referring to the Tcf1⁺ cells that overlaps with the peripheral Tcf1⁻ population. Unfortunately we cannot track which sequences are responding to viral reactivation events, but based on statistics it seems that the higher the abundance of a specific Tcf1⁺ clone, the more chance it has to encounter antigen and give rise to the Tcf1⁻ cells. The M38-specific clonotypes that do not make it into the peripheral pool, could be of a lower affinity, as we have recently shown that mainly high avidity T cells are recruited into the inflationary T cells within the first 100 days post infection [10]. However, as we have only screened the TCR Vβ chain, we do not know the pairing of each clone with the TCR Vα chain. This prevents us from cloning the TCRs and determining the avidities of differentially recruited clones.

We suspect, if these Tcf1⁺ M38-specific cells at some point would encounter antigen they would respond by proliferation and would also give rise to Tcf1⁻ cells. If these cells do not see antigen, they are maintained by homeostatic proliferation and maintain Tcf1 expression.

References:

- 1 **Manesso, E., Kueh, H. Y., Freedman, G., Rothenberg, E. V. and Peterson, C.,** Irreversibility of T-Cell Specification: Insights from Computational Modelling of a Minimal Network Architecture. *PLoS One* 2016. **11**: e0161260.

- 2 **Corish, P. and Tyler-Smith, C.**, Attenuation of green fluorescent protein half-life in mammalian cells. *Protein Eng* 1999. **12**: 1035-1040.
- 3 **Welten, S. P., Redeker, A., Franken, K. L., Benedict, C. A., Yagita, H., Wensveen, F. M., Borst, J. et al.**, CD27-CD70 costimulation controls T cell immunity during acute and persistent cytomegalovirus infection. *J Virol* 2013. **87**: 6851-6865.
- 4 **Remmerswaal, E. B., Klarenbeek, P. L., Alves, N. L., Doorenspleet, M. E., van Schaik, B. D., Esveldt, R. E., Idu, M. M. et al.**, Clonal evolution of CD8+ T cell responses against latent viruses: relationship among phenotype, localization, and function. *J Virol* 2015. **89**: 568-580.
- 5 **Gordon, C. L., Miron, M., Thome, J. J. C., Matsuoka, N., Weiner, J., Rak, M. A., Igarashi, S. et al.**, Tissue reservoirs of antiviral T cell immunity in persistent human CMV infection. *Journal of Experimental Medicine* 2017. **214**: 651-667.
- 6 **Miron, M., Kumar, B. V., Meng, W. Z., Granot, T., Carpenter, D. J., Senda, T., Chen, D. et al.**, Human Lymph Nodes Maintain TCF-1(hi) Memory T Cells with High Functional Potential and Clonal Diversity throughout Life. *Journal of Immunology* 2018. **201**: 2132-2140.
- 7 **Kratchmarov, R., Magun, A. M. and Reiner, S. L.**, TCF1 expression marks self-renewing human CD8(+) T cells. *Blood Adv* 2018. **2**: 1685-1690.
- 8 **Wieland, D., Kemming, J., Schuch, A., Emmerich, F., Knolle, P., Neumann-Haefelin, C., Held, W. et al.**, TCF1(+) hepatitis C virus-specific CD8(+) T cells are maintained after cessation of chronic antigen stimulation. *Nature Communications* 2017. **8**.
- 9 **Welten, S. P., Redeker, A., Franken, K. L., Oduro, J. D., Ossendorp, F., Cicin-Sain, L., Melief, C. J. et al.**, The viral context instructs the redundancy of costimulatory pathways in driving CD8(+) T cell expansion. *Elife* 2015. **4**.
- 10 **Baumann, N. S., Welten, S. P. M., Torti, N., Pallmer, K., Borsa, M., Barnstorf, I., Oduro, J. D. et al.**, Early primed KLRG1- CMV-specific T cells determine the size of the inflationary T cell pool. *PLoS Pathog* 2019. **15**: e1007785.
- 11 **Welten, S. P. M., Baumann, N. S. and Oxenius, A.**, Fuel and brake of memory T cell inflation. *Med Microbiol Immunol* 2019. **208**: 329-338.
- 12 **Redeker, A., Welten, S. P. and Arens, R.**, Viral inoculum dose impacts memory T-cell inflation. *Eur J Immunol* 2014. **44**: 1046-1057.
- 13 **Snyder, C. M., Cho, K. S., Bonnett, E. L., van Dommelen, S., Shellam, G. R. and Hill, A. B.**, Memory inflation during chronic viral infection is maintained by continuous production of short-lived, functional T cells. *Immunity* 2008. **29**: 650-659.
- 14 **Jeannet, G., Boudousquie, C., Gardiol, N., Kang, J., Huelsken, J. and Held, W.**, Essential role of the Wnt pathway effector Tcf-1 for the establishment of functional CD8 T cell memory. *Proc Natl Acad Sci U S A* 2010. **107**: 9777-9782.
- 15 **Zhou, X., Yu, S., Zhao, D. M., Harty, J. T., Badovinac, V. P. and Xue, H. H.**, Differentiation and persistence of memory CD8(+) T cells depend on T cell factor 1. *Immunity* 2010. **33**: 229-240.

REVIEWERS' COMMENTS:

Reviewer #1 (Remarks to the Author):

The authors have adequately addressed the reviewers' concerns.

Reviewer #2 (Remarks to the Author):

The authors did a great job in revising an already strong manuscript. All my questions were sufficiently answered.

Reviewer #3 (Remarks to the Author):

I would like to thank the author for their detailed replies to my comments. I think that they have done an excellent job with the clarifications and revisions and I am very satisfied with the manuscript.